# Causal Parrots: Large Language Models May Talk Causality But Are Not Causal

**Matej Zečević\***                                                          *matej.zecevic@tu-darmstadt.de*
*Computer Science Department, TU Darmstadt, Germany*

**Moritz Willig\***                                                          *moritz.willig@cs.tu-darmstadt.de*
*Computer Science Department, TU Darmstadt, Germany*

**Devendra Singh Dhami**                                                      *devendra.dhami@tu-darmstadt.de*
*Computer Science Department, TU Darmstadt, Germany*
*Hessian Center for AI (hessian.AI), Germany*

**Kristian Kersting**                                                         *kersting@cs.tu-darmstadt.de*
*Computer Science Department, TU Darmstadt, Germany*
*Centre for Cognitive Science, TU Darmstadt, Germany*
*Hessian Center for AI (hessian.AI), Germany*
*German Research Center for Artificial Intelligence (DFKI), Germany*

**Reviewed on OpenReview:** *https://openreview.net/forum?id=tv46tCzs83*

## Abstract

Some argue scale is all what is needed to achieve AI, covering even causal models. We make it clear that large language models (LLMs) cannot be causal and give reason onto why sometimes we might feel otherwise. To this end, we define and exemplify a new subgroup of Structural Causal Model (SCM) that we call meta SCM which encode causal facts about other SCM within their variables. We conjecture that in the cases where LLM succeed in doing causal inference, underlying was a respective meta SCM that exposed correlations between causal facts in natural language on whose data the LLM was ultimately trained. If our hypothesis holds true, then this would imply that LLMs are like parrots in that they simply recite the causal knowledge embedded in the data. Our empirical analysis provides favoring evidence that current LLMs are even weak 'causal parrots.'

## 1 Introduction

Speaking of causality, the Pearlian counterfactual theory of causation has recently found prominent support in the AI/ML community (Schölkopf, 2022; Peters et al., 2017; Geffner et al., 2022). An increasing presence of publications at major conferences/journals concerned with the integration of causality with AI/ML (including (Janzing & Schölkopf, 2018; Lee & Bareinboim, 2019; Löwe et al., 2022; McDuff et al., 2022; Zečević et al., 2021; Lu et al., 2022) to mention a select few) suggests a growing subfield that sets a consensus on *causal* AI/ML as promising paradigm for next-generation systems. Still, as the difficulty of the integration with otherwise prominent success stories of deep learning, such as computer vision, becomes apparent, countering opinions start to speak out against causal AI/ML (Bishop, 2021). In this work, we take the former perspective *pro* causal AI/ML. We argue that the questions around causality can fuel research also on questions of recent debates such as how much 'real' progress towards AGI has been made since the advent of large scale models such as BERT (Devlin et al., 2018), GPT-3 (Brown et al., 2020), DALL-E (Ramesh et al., 2021).

The following block paragraph serves as a summary of an example of such a recent debate on whether 'just scaling' these models is a sufficient condition for progression towards AGI:

---

\* Shared first co-authorship.

> With the rise of large scale models such as BERT, GPT-3, DALL-E, AI history suggests to repeat itself as arguably impressive text generation and image synthesis results foster different opinions in the community in terms of interpretation regarding the progression of the field as a whole towards the grand goal of AGI (key references involve (Marcus & Davis, 2021; Marcus, 2022) that sparked intense discussions amongst notable researchers via social networks; also for reference, a short treatise that discussed patterns in the history of AI research observes: "early, dramatic success followed by sudden unexpected difficulties." (Chauvet, 2018)). Some even speak of *foundation* models (Bommasani et al., 2021) to account for the "emerging paradigm" of models that provide a base from which task-specific models are derived through adaptation. The emergence of what seem to be the two dominant clusters of opinions within the discussion around foundation models is characterized by researchers who recognize said models as significant progression towards AGI and those who do *not*. The former group of the observed results act as corroborating evidence for the *scaling hypothesis* (Branwen, 2020; Sutton, 2019) which captures that the idea of emergent properties as a result of scaling neural network in terms of parameters and data, thereby, rooting parts of the overarching idea in results from neuroscience that suggest the human brain to 'just' be a scaled up primate brain (Herculano-Houzel, 2012). The other popular opinion is that the achieved results act as a mere reflection of the sheer scale of data and parameters, put differently "the methods are old" and their lack of interpretability and reasoning capabilities will remain persistent. While many researchers voiced their opinions and thoughts, one notable comment came from Judea Pearl who announced his alliance with the latter position via social media, stating "These models are castles in the air. They have no foundations whatsoever." discrediting the models for lacking any identifiable notion to causality.

It is clear how resolving these discussion through scientific inquiry is crucial for the AI/ML community. The question of the whether or not Pearl's statement is true was one of the seeds that grew in to the present paper.

*Therefore, in this paper, we investigate whether current foundation models are such "castles in the air."*

We identify the key problem of ongoing debates to lie in the scale of data and parameters that only further cement the inherently black-box nature of the base models. Therefore, to answer whether such "foundation models" have made progress towards AGI and to give reason onto why causal AI/ML could be a milestone, it seems to suffice to ask and investigate the question of the extent to which *foundation models can talk causality.* For the sake of simplicity, we will take the "talking" literally and focus on LLMs in this work while leaving general foundation models (e.g. image-based models) for future work.[1]

The paper is structured as follows: we investigate how 'causal' LLMs are by first formalizing our key hypothesis on "correlations of causal facts" using Pearl's language, introducing necessary new ideas along the way with illustrative examples, posing our key theoretical contribution. Following that, we provide an empirical analysis on the causal prowess of current LLMs and discuss the results in lights of our theoretical groundwork from the beginning, posing our second contribution.

For reproduction purposes we make our code repository for the empirical part publicly available.[2]

## 2 Informal Summary of the Main Idea of the Paper

LLMs are transformer-based models (Vaswani et al., 2017) and it is clear how they are *not* parameterized variants of SCMs such as the neural ones presented in (Xia et al., 2021) as they do not specify structural equations mimicking causal mechanisms. Nonetheless, they do seem to answer causal questions right sometimes. Our explanation for this is that they are not only 'stochastic parrots' as already suggested by Bender et al. (2021) but sometimes also 'causal parrots' since they will also encounter correlations over causal facts during training in their vast oceans of textual data. Essentially, LLMs can be trained to produce arbitrary text of desired content. In the case of this paper 'desired' means to produce the right causal answers. We

---

[1]However, for observations that we believe should also hold more generally for foundation models (like the concept of "correlations of causal facts" that we will introduce in this work) we will use the term foundation model instead of writing LLM.

[2]https://github.com/MoritzWillig/causalParrots/

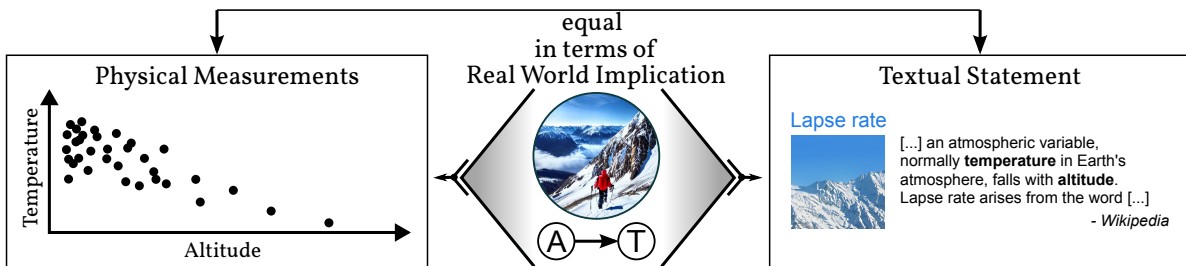

Figure 1: **Same Implication, Different Representations.** When we consider the causal relationship between altitude (A) and temperature (T), then it is apparent that given the laws of physics we have an increase in altitude leading to a decrease in temperature. Graphically we can depict the relationship as A→T, whereas the actual 'increase-decrease' relationship can only be specified through the SCM formalism with its structural equations, that is some $f$ such that $T = f(A, U)$ where $U$ are exogenous variables. The ground truth SCM underlying our laws of physics generates observational data in the form of numerical tuples $(a, t)$ as seen on the left scatter plot. To infer the casual relation, we can resort to algorithms for causal discovery. However, crucially, the same knowledge achieved through such induction can be represented within *text* for 'free' as one simply recites the Wikipedia article found on the right. While the article on the right is correct, and thus represents a fact about the actual world, there is no such guarantee for arbitrary other texts. That is, a model that simply obtains its knowledge from various Wikipedia statements will also learn untrue statements, statements that are not facts, thus explaining behavior that is correct sometimes and wrong other times.

illustrate this idea in Fig.1 where we show the example of the physical concept that we know as altitude and how it is being causal of the concept of temperature–in the sense that there is a physical mechanism that leads to a temperature increase/decrease with a corresponding change in altitude–is a fact of causality in our physical reality that can be represented and therefore also learned in different ways. One way, the way that we usually consider in ML, is through induction on physical measurements. We have actual data points of different altitude-temperature pairs (maybe recorded from different cities and mountain ranges) and infer the underlying relationship. For this we could fit a linear causal model with non-gaussian noise (LiNGAM; see the original work for reference (Shimizu et al., 2006)) and see that it explains our data perfectly, concluding that altitude causes temperature. However, this conclusion is arguably nothing new, as most people would agree, and this is partly so because such obtained knowledge has been embedded as textual articles into encyclopedias such as Wikipedia, which are freely accessible. It turns out that not just humans but also LLMs learn from precisely such textual data that already contain the results from the original physics experiments with 'real' data. For instance, 'The Pile' data set which was used for training OPT (Zhang et al., 2022b) comprised over 6 GiB of textual data from Wikipedia (Gao et al., 2020). Both physical measurements and the facts we find on Wikipedia are a consequence of the causal reality that altitude causally influences temperature and reversely they both imply this same causal model. However, they are fundamentally different forms of representation in that learning from the physical measurements can be argued to be what we mean by 'understanding', whereas simply reading up on the textual article lacks exactly that component. Indeed, this discussion then turns philosophical onto which we will add a few more comments.

While the philosophical argument introduced by Searle (2009) is essentially the same as the stochastic/causal parrot argument without the probabilistic view, another compelling comparison of LLMs is that of the Plato's allegory of the cave (Plato, 375 BC) in which the question is raised to which extent one can learn about the real world's functioning by just observing the shadows of its objects. For this allegory, we could see the LLM act as the cave where one can observe some causality in the form of correct answers (the 'shadows') but the question is raised whether this would in fact be actual *causality* (the 'real world'). Even if we consider LLMs to be universal in the same sense that we have proven that neural networks are universal function approximators (Cybenko, 1989) that does not imply that it is easy to make them causal and therefore even if they can make use of correlations exposed by meta SCM talking about causal facts, then still the LLM would be required to be exposed to an infinite amount of data from such universal, meta SCM.

## 3 Formalizing "Correlations of Causal Facts"

"Correlation does not imply causation," goes the famous saying (see Aldrich (1995); Pearl (2009)), that accounts for the fact that following Reichenbach's *common cause principle* a correlation between two variables might be either because one is causing the other, or because there is a third variable causing both (Reichenbach, 1956). To infer the 'actual' causation[3] within the system of interest, we might resort to *manipulating* the system, as another famous saying suggests "No causation without manipulation" (Holland, 1986). A celebrated victory of Pearl's notion to causality is the *causal hiearchy theorem* (CHT) which guarantees that purely observational data collected from a system can not be used to uniquely determine causal statements, when no other causal assumptions are available (Bareinboim et al., 2020). The CHT certainly seems to imply that *no matter how much* we scale our foundation models (in terms of data and parameters), we will never be able to perform causal inference. In a nutshell, the CHT seems to disprove the scaling hypothesis. Or does it? In this work, we argue that foundation models might be exploiting a "loop hole" in the CHT[4]. Namely, what happens if the *causal assumptions* (which are required, by the CHT, for causal inference) are represented in observational data itself?

We start by providing the definition of a Structural Causal Model (SCM). We will follow the broadest notion of SCM that still enjoys properties such as unique solvability, marginalization and an intuitive graphical underpinning. The class is that of *simple* SCM as first introduced by Bongers et al. (2021). They extend typical acyclic, semi-Markovian[5] SCM with cycles that offer a unique solution[6]. Formally, we have:

**Definition 1.** *A simple Structural Causal Model (SCM) is a tuple* $\mathcal{M} := (\mathcal{I}, \mathcal{J}, \boldsymbol{\mathcal{X}}, \boldsymbol{\mathcal{E}}, \boldsymbol{f}, \mathbb{P}_{\boldsymbol{\mathcal{E}}})$ *where* $\mathcal{I}, \mathcal{J}$ *are disjoint, finite index sets[7] for endo- and exogenous variables respectively,* $\boldsymbol{\mathcal{X}} = \prod_{i \in \mathcal{I}} \mathcal{X}_i, \boldsymbol{\mathcal{E}} = \prod_{j \in \mathcal{J}} \mathcal{E}_i$ *are products of the domains of the variables where each domain is a standard measurable space[8],* $\boldsymbol{f}$ *are the structural equations for which* $\mathbf{X} = \boldsymbol{f}(\mathbf{X}, \mathbf{E})$ *for random variables* $\mathbf{X}, \mathbf{E}$, *and finally* $\mathbb{P}_{\boldsymbol{\mathcal{E}}}$ *denotes the exogenous distribution.* $\mathcal{M}$ *is further uniquely solvable[9] for every subset* $\mathcal{O} \subseteq \mathcal{I}$. *The graph implied by* $\mathcal{M}$ *is denoted as* $\mathcal{G}(\mathcal{M})$. ∎

The graph of a simple SCM is drawn by considering the parent-child relationships implied by the structural equations. We call a variable $k$ *parent* of some variable $i$ if there exists no measurable function that can match the actual structural equation that computes the values of $i$ while not using $k$ as argument. For simple SCM we then end up having three basic types of *causal* relationships that can exist for any variable pair $(i, j)$ of $\mathcal{G}(\mathcal{M})$: (1) if there is a direct edge $i \rightarrow j \in \mathcal{G}(\mathcal{M})$, then $i$ is called a direct cause of $j$, (2) if there is a directed path $i \rightarrow \cdots \rightarrow j \in \mathcal{G}(\mathcal{M})$, then $i$ is simply a cause, and (3) if there is a bidirected edge $i \leftrightarrow j \in \mathcal{G}(\mathcal{M})$, then $i$ and $j$ are confounded. In the following, we will provide an example of a simple SCM based on the 'classical' setting described in Fig.1 using the above definition.

**Example 1** ('Classical Setting'). Let $X, Y, Z$ be the random variables whose values describe "Country's Annual Per Capita Chocolate Consumption", "Number of Nobel Laureates per 10 Million Population" and the "Gross Domestic Product" for any given country respectively. The data observed in Messerli (2012) suggested a significant linear correlation ($r = 0.791, p < 0.0001$) for $(X, Y)$ with Switzerland being the top-performer ($X > 10, Y > 30$). Note that $Z$ was not observed but clearly serves as a reasonable explanation for the underlying causal system. As postulated by Reichenbach's common cause principle (Reichenbach, 1956), if the correlation does not imply a direct causation, then there must exist a common cause, which we'd choose as $Z$ for this particular example. While we do not know the true SCM accountable for

---

[3]For a rigorous treatment of different notions of what consitutes an "actual cause" consider the seminal work of Halpern (2016).

[4]Or rather, it is a *subtle* detail that might easily be forgotten.

[5]SCMs that allow for latent confounding, that is, the exogenous terms need not be independent.

[6]However, these do unfortunately exclude self cycles and many other circular relationships.

[7]For examples later on, we might simply equate indices to letters when considered more suitable for presentation, that is, instead of having $\mathcal{I} := \mathbf{3} = \{1, 2, 3\}$ we write $\mathcal{I} := \{X, Y, Z\}$.

[8]See Def.F.1 of (Bongers et al., 2021) that gives a rigorous, albeit very technical definition. Typically, definitions on SCMs found in the literature are rather hand-wavy about what the 'variables' are. For instance Bareinboim et al. (2022) (Def.1) simply refers to 'variables' in the most general sense, while Pearl (2009) is more specific in talking about 'random variables'. However, probability theorists often restrict themselves to separable completely metrizable spaces also known as *Polish spaces*, which avoid pathological situations in which the considered spaces' sizes are uncountable, that is, greater/equal $2^{\aleph_0}$.

[9]See Def.3.1. in (Bongers et al., 2021).

$X, Y, Z$, we can approximate the data from Messerli (2012) reasonably well using the following simple SCM $\mathcal{M}_1 := (\{X, Y, Z\}, \mathbf{3}, \mathbb{R}^3, \mathbb{R}^3, \boldsymbol{f}, \mathbb{P}_{\mathbb{R}^3})$ with $\boldsymbol{f}$ being defined as

$$\boldsymbol{f} := \{X = f_1(E_1) = E_1, \quad Y = f_2(X, E_2) = 2 \cdot X + E_2, \quad Z = f_3(E_3) = E_3\}^{10}. \tag{1}$$

The structural equations in (1) make $\mathcal{M}_1$ a linear additive noise model that approximates the data from Messerli (2012) reasonably well. It is clear how the observed correlation in this case corresponds to a direct causation according to $\boldsymbol{f}$ since $X$ is a parent of $Y$ and $Z$ simply an independent variable. However, we are certain that chocolate consumption does not cause more Nobel laureates anywhere, so $\mathcal{M}_1$ is not reasonable in predicting our real world expectations. Following Reichenbach's postulate, it would be more reasonable to use an alternate SCM $\mathcal{M}_2 := (\{X, Y\}, \mathbf{3}, \mathbb{R}^2, \mathbb{R}^3, \boldsymbol{f'}, \mathbb{P}_{\mathbb{R}^3})$ with $\boldsymbol{f'}$ being defined as

$$\boldsymbol{f'} := \{X = f_1'(E_1, E_3) = E_3 + E_1, \quad Y = f_2'(E_2, E_3) = 2 \cdot E_3 + E_2\}. \tag{2}$$

This second SCM $\mathcal{M}_2$ would now correspond better to both (1) the actual data observed in (Messerli, 2012) since $Z$ was never observed and is modelled as exogenous variable $E_3$ implicitly and (2) to our real world intuition since there is a bidirected edge $X \leftrightarrow Y \in \mathcal{G}(\mathcal{M}_2)$ with $E_3$ being the underlying confounder. ∎

Example 1 serves to show how the rather abstract definition of an SCM can be made tangible to communicate what we believe about our observed data and more so the underlying data generating process. Previously, we have defined a (direct) cause as a directed (1-)path in the causal graph, however, we have not discussed *why* we call such an edge in the implied graph of an SCM a 'cause'. The reasoning behind the naming turns out is an important insight that we will use soon to develop our idea of "correlation of causal facts." But first, we need to briefly talk about Pearl's Causal Hierarchy (PCH) which defines three (symbolic) languages $\mathcal{L}_1, \mathcal{L}_2, \mathcal{L}_3$ with increasingly expressive quantities. For this, we make use of the definitions proposed by Bareinboim et al. (2022).

**Definition 2.** *The Pearl's Causal Hierarchy (PCH) consists of three (symbolic) languages $\mathcal{L}_1, \mathcal{L}_2, \mathcal{L}_3$. The terms are called: (1) observational $P(\mathbf{Y} = \mathbf{y}) \in \mathcal{L}_1$[11], (2) interventional $P(\mathbf{Y_x} = \mathbf{y}) \in \mathcal{L}_2$, and (3) counterfactual $P(\mathbf{Y_x} = \mathbf{y}, \dots, \mathbf{Z_w} = \mathbf{z}) \in \mathcal{L}_3$. Where $P(\mathbf{Y_x} = \mathbf{y})$ denotes the probability of $\mathbf{Y}$ being $\mathbf{y}$ were $\mathbf{X}$ to have values $\mathbf{x}$*[12]. ∎

Now we can return to clarifying why we call certain paths in the causal graph 'causal'. We state our insight as:

**Insight 1.** *Let $\mathcal{M}$ be some SCM. Knowledge about the structural equations and the causal graph of $\mathcal{M}$ is knowledge about answering $\mathcal{L}_3$ and $\mathcal{L}_2$ queries in $\mathcal{M}$ respectively.* ∎

Returning to Example 1, it is clear how knowing the actual parameterizations of the functions $f_1', f_2', f_3'$ allows us to answer $\mathcal{L}_3$ queries and not knowing the actual parameterizations but at least knowing that for instance variable $Z$ is a parent of $X$ and therefore a necessary argument in $f_1'$ is sufficient for answering $\mathcal{L}_2$ queries. Put differently, we can call a direct edge $Z \to X \in \mathcal{G}(\mathcal{M}_2)$ 'causal' since it is a $\mathcal{L}_2$ fact[13]. We can rephrase this statement differently to make clear where our overall idea is leading to. Namely, some machine learning model is 'causal' w.r.t. some query if it can answer that query with the right $\mathcal{L}_2$ fact. More intriguingly, it does not matter where that $\mathcal{L}_2$ fact comes from since the formulation is independent of whether the model *learns* the fact and simply requires that the model *knows* about the fact. We state our second key insight as:

**Insight 2.** *The 'variables' of SCMs are not restricted to 'natural' concepts such as "Chocolate consumption" or "Number of Nobel laureates" (see Ex.1), they can be 'meta' concepts involving causal facts, that is, knowledge about $\mathcal{L}_2$ and $\mathcal{L}_3$.* ∎

---

[10]Note how these equations make the assumption that all the variables are measured using the *same scale*. This is of course not a necessary assumption, however, since this example serves only illustrative purposes for clarifying the ideas around SCMs, and we do not train an actual SCM on the data from (Messerli, 2012), the chosen presentation is reasonable.

[11]We use $P(\mathbf{X} = \mathbf{x})$ to simply denote the probability value of random variable $\mathbf{X}$ obtaining value $\mathbf{x}$.

[12]Not to be confused with regular conditionals $P(\mathbf{Y} = \mathbf{y} \mid \mathbf{X} = \mathbf{x}) \in \mathcal{L}_1$ that are read as "the probability of $\mathbf{Y}$ being $\mathbf{y}$ were $\mathbf{X}$ *observed* as $\mathbf{x}$".

[13]How we actually acquire such $\mathcal{L}_2$ (or even $\mathcal{L}_3$) facts is a different question altogether. It stands at the heart of the discipline of *causal discovery*. We call $\mathcal{L}_2$ 'interventional' since we can use experimentation in the form of 'surgical intervention' on admissible variables measured in the given task to discern causation from mere correlation.

For our argument the existence of data describing $\mathcal{L}_2$ facts is already sufficient, therefore, for ease of presentation we will focus on those specifically in the following although we see no difficulties (as of writing) in extending to $\mathcal{L}_3$. We call these concepts 'meta' since they are one level above 'regular', simple SCM in the sense that they *encode information about answering causal questions in another SCM*. To make this idea more formal, we define 'meta' SCM as follows:

**Definition 3.** *Let $\mathcal{M}_1$ and $\mathcal{M}_2$ be two SCMs such that the observational distribution of $\mathcal{M}_2$ denoted $\mathcal{L}_1(\mathcal{M}_2)$ can answer queries w.r.t. the interventional distributions of $\mathcal{M}_1$ denoted $\mathcal{L}_2(\mathcal{M}_1)$, then $\mathcal{M}_2$ is called meta (w.r.t. $\mathcal{M}_1$).* ∎

In the following, we construct an example of such a meta SCM analogue to the 'classical setting' from Example 1.

**Example 2** ('Meta Setting'). As before, let $X, Y, Z$ denote the natural concept random variables from Example 1 for the chocolate-Nobel data from Messerli (2012) and set $\mathcal{M}_1 := (\{X, Y\}, \mathbf{3}, \{0, 1\}^2, \{0, 1\}^3, \boldsymbol{f}, \mathbb{P}_{\{0,1\}^3})$ with $\boldsymbol{f} := \{X = f_1(E_1, E_3) = E_3 \wedge E_1, Y = f_2(E_2, E_3) = E_3 \wedge E_2\}$ where $X = 1, Y = 1$ denote 'high' chocolate consumption and number of Nobel laureates respectively, and vice versa for $X = 0, Y = 0$[14]. In this example, we intend on answering an arbitrary example query from $\mathcal{L}_2(\mathcal{M}_1)$ like for instance $P(Y_{X \leftarrow 1} = 1)$, that is, the probability of a high number of Nobel laureates if the given chocolate consumption were to be high. Clearly, since $X \leftrightarrow Y \in \mathcal{G}(\mathcal{M}_1)$, we expect $P(Y_{X \leftarrow 1} = 1) = P(Y = 1)$ since intervening on $X$ will not change $Y$. For the sake of the example let's assume fair coinflips as exogenous distributions, $\mathbb{P}_{\mathcal{E}_i} := \mathcal{B}(1/2)$, then $P(Y_{X \leftarrow 1} = 1) = 1/4$. Next, we need to show the existence of some SCM $\mathcal{M}_2$ that is meta to $\mathcal{M}_1$, that is, which will answer $P(Y_{X \leftarrow 1} = 1)$ using $\mathcal{L}_1(\mathcal{M}_2)$. This is easy, as we can define

$$\mathcal{M}_2 := (\{W\}, \emptyset, \{0, 1\}^{3 \times 3}, \emptyset, \boldsymbol{f}', \mathbb{P}_\emptyset) \tag{3}$$

where $W := \mathcal{G}(\mathcal{M}_1)$ is a random variable that describes a causal graph over $X, Y, Z$ as an adjacency matrix[15] and therefore

$$\boldsymbol{f}' := \{W = f_1'(W) = \left(\begin{smallmatrix} 0 & 0 & 0 \\ 0 & 0 & 0 \\ 1 & 1 & 0 \end{smallmatrix}\right)\}. \tag{4}$$

$W$ encodes the causal graph $X \leftarrow Z \rightarrow Y$. It is now a simple exercise to show that $\mathcal{M}_2$ is indeed meta. We want to find the $a$ in $P(Y_{X \leftarrow 1} = 1) = a$. We know that $P(Y = 1) = 1/4$ and using $\mathcal{L}_1(\mathcal{M}_2) = W$ we further know that $X$ is **not** a parent of $Y$ since $X \rightarrow Y \notin W = \mathcal{G}(\mathcal{M}_1)$. Therefore, we conclude that $P(Y_{X \leftarrow 1} = 1) = P(Y = 1) = 1/4 = a$. ∎

At this point, it is worthwhile noting that while every SCM is in fact a causal model it is not the case that every SCM is a reasonable causal model for our physical world or rather for what is being modelled. While the meta SCM in the above case can also be viewed even as a simple SCM which retains even its causal interpretation on graphs, it is surely not a 'regular' SCM particularly because it is in direct reference of another SCM. We have achieved two things in the above example (1) shown the existence of meta SCM and (2) shown how they can actually answer concrete causal facts about another SCM. While $\mathcal{M}_2$ from the above example may seem artificial in the sense that it is tailored to $\mathcal{M}_1$, the example acts as intended since we could clearly define a suitable meta SCM for our illustration's purposes.[16] After having proven the existence of meta SCM that can encode $\mathcal{L}_2$ facts (of another SCM) within their observational distribution, hence making them meta, we are finally ready to provide our main running hypothesis for this work that we believe to hold, namely *correlations of causal facts*.

**Conjecture 1** (Correlation of Causal Facts (CCF)). *As before, let $\mathcal{M}_1$ be some SCM and $\mathcal{M}_2$ a respective meta SCM. Further let $Q \subset \mathcal{L}_2(\mathcal{M}_1)$ and $A \subset \mathcal{L}_1(\mathcal{M}_2)$ be causal queries with their respective answers and $f$ denotes the LLM's predictive model. Then we have: $f(Q) = A \iff f(Q)$ minimizes training error.* ∎

In words, Conj.1 suggests that in all the cases where the LLM does provide the right causal answer to a causal query, then it is only because (i) this fact was observed in the training data and (ii) the correlation with the query is optimal from the perspective of the training objective. This formulation also leaves room

---

[14]We adapted the SCM to be binary for ease of calculation in the example, the definition holds w.l.o.g. for any simple SCM.
[15]A cell $A_{ij} = 1$ indicates the direct edge $i \rightarrow j$.
[16]We could have chosen $W$ to represent the space of all directed acyclic graphs (DAG) and eventually found the same conclusion, albeit at the cost of clarity for the example.

for the fact that LLMs being right will not always (and maybe not even most of the times) be the case. Furthermore, it provides another very intriguing observation in that an LLM, although we conjecture it to essentially be involved with at least two SCM (the one modelling the phenomenon of interest and the meta one modelling the first model) during training, it is not causal and cannot be (up to answering some subset of causal questions right). To prove our conjecture, we would require three things: (i) identify meta SCM that generated the causal facts in the LLMs training data, (ii) show that the causal facts are the best (in terms of training loss) answers to the causal queries and (iii) show that LLMs indeed give the causal facts as answers to the causal question. Since (i-ii) are arguably infeasible or at least fiendishly difficult to validate, in the following, we will focus our initial empirical analysis on (iii) and judge the causal inference capabilities of LLMs.

**Meta SCM and Fine-Tuning of General LLMs.** As discussed in (Bommasani et al., 2021), to optimize general-purpose LLMs (like for instance GPT) for any particular downstream-task, fine-tuning is an appropriate method for doing so. When considering the just presented formalism around meta SCM, then a natural question to ask is to which extent we can consider the meta SCM to be a 'reliable' proxy for the downstream application to be solved. The term 'reliable' here can be understood as 'correct' in the sense of being a meta SCM that describes the properties of the true underlying data-generating process for the downstream task. The research question then amounts to answering *whether fine-tuning simply means to find the right meta SCM for the downstream task.* While orthogonal at first to the problem solved in this particular work, it is an intriguing question that so far only allows us to provide some educational guess. We posit that the answer to the previous question is affirmative in that fine-tuning data sets will be chosen in accordance to the downstream task, however crucially, the LLM architecture/paradigm is not being changed. Therefore, the CCF idea still applies to our fine-tuned LLM in that now the correlations of causal facts are found in the fine-tuning training data.

## 4 Testing for Causal Knowledge in Large Language Models

We evaluate three publicly accessible LLMs: OpenAI's GPT-3 ('text-similarity-davinci-001', Brown et al. (2020)), AlephAlpha's Luminous ('luminous-base', AlephAlpha (2022)), and Meta's OPT ('OPT-30B', Zhang et al. (2022b)). All models are transformer based architectures (Vaswani et al., 2017) trained at scale, qualifying them as LLMs (see Bommasani et al. (2021)). Our analysis primarily investigates three different questions that belong to part (iii) of CCF as described earlier, namely:

*How do LLMs perform..*

> *..in "common sense" settings like reasoning or intuitive physics?*
> *..in settings where the causal graph is (partially) known?*
> *..when using their embeddings of knowledge base facts?*

In the following, we conduct various experiments to answer these questions.

### 4.1 Methods and Results for "Common Sense" Inference Tasks

We argue that "common sense" reasoning tasks that either involve some basic propositional logic or intuitive physics (as reference consider for instance Tenenbaum et al. (2011)) are reasonable settings in which we can expect CCF to hold. For propositional logic we consider 20 different questions such as for example "If $A$ causes $B$ and $B$ causes $C$ does $A$ cause $C$?". These questions are simply fed as prompt to the respective LLM. In this setup no prior world knowledge is required, other than being able to link together propositions. In the simplest propositional chain we provide the model with a chain of three variables, linked by two propositions: "$A$ causes $B$", "$B$ causes $C$." We ask the model "Does $A$ cause $C$?". This setup provides a fairly simple and easy to control experiment. We vary the length of the chain up to $n = 10$ variables: "$X_1$ causes $X_2$ ... and $X_{n-1}$ causes $X_n$". (Note that upon instantiation of the queries placeholders are replaced with the corresponding letters: $X_1 \rightarrow$'A', $X_2 \rightarrow$'B', ...). To prevent the model from simply pattern matching the first and last variable we ask for sub-chains (e.g. "Does $B$ cause $E$?") that leave out the initial or last part of the chain. Additionally, we might also switch the order of propositions or exchange the alphabetical ordered variable names with random letters.

Results for this experiment are shown in Table 1. A complete list of questions and answers can be found in the appendix. We observe that GPT-3 can handle shorter chains up to 5 and starts to fail at $n = 6$ variables. Contrary to this, Luminous start to answer correctly at $n = 6$ variables but fails at shorter chains. OPT only answers two queries correctly. For most of our evaluations we observe the tendency of OPT to repeat the given query text and not answering the prompt. For all three models we see a decline in performance once we leave the standard $X_1...X_n$ setting and start to query for sub-chains and randomize proposition order. Only Luminous keeps reasonable performance for the 'Randomized' queries. From our observations we conclude that current models still need special fine tuning such as in (Zhang et al., 2022a) to be applicable to inference tasks with given information.

Since LLMs usually operate on natural language, performance on Causal Chains might be reduced by the use of symbolic variable names ("If $A$ causes $B$, ..."). Thus, a simple additional Natural Word Chain experiment is performed to test whether model performance on causal chains is due to the use of single letter variable names or not. The 'Natural Word Chain' experiment tests LLM performance on simple causal 2-chains $(X \rightarrow Y \rightarrow Z)$ with varying variable names. Unconnected chains $(X \rightarrow Y, W \rightarrow Z)$ are prompted to test for non-causal connections. The 'Real World' subset tests the causal chain of 'temperatures rising' $\rightarrow$ 'poles melting' $\rightarrow$ 'sea level rising'. With any of the variables being replaced by 'people walking faster' to test for causal inference with non-factual information. The 'Imaginary' subset presents causal chains with the non-existing but natural sounding concepts 'Quab', 'Blaong' and 'Wahrg'. The 'Mixed' subset combines variable names of both previous subsets. No significant improvement or decrease in performance was observed in any of the models. GPT-4 with CoT prompting manages to correctly answer all questions by strictly adhering to the provided information only. Thus, avoiding reasoning about any assumed external information that leads to previously wrong conclusions in the non-CoT case.

For intuitive physics we consider 36 questions such as for example "A ball is placed on a table and rolls off. What does this tell us about the table?". As we are not able to feed images depicting the physical scenes into the LLMs, we are resorting to textual descriptions. We provide idealized descriptions of scenes that query the understanding about physical mechanisms, such as balls rolling off tilted surfaces or objects bouncing away in a collision. Using textual descriptions leads to ambiguity as weights, sizes and relative positions of the objects might not be described. To compensate for this uncertainty, we choose simple setups that might even occur in text books and count all *plausible* outcomes given by the LLMs (as manually evaluated by us) as correct.

Again, results for this experiment are shown in Table 1. The most striking observation that caught us were some parallels to *human* nature of reasoning. In the 'Weights' cluster of questions we asked the LLMs to answer variations of the well-known trick question "What is heavier: A kilogram of metal or a kilogram of feathers?". For this questions all models[17] wrongly opt for the material with higher specific weight and answer "A kilogram of metal is heavier than a kilogram of feathers." However, when asked "['A kilogram of metal is heavier than a kilogram of feathers'] is what most people say, but in reality", GPT-3 correctly answers "They weigh the same." arguably the same way many humans intuitively do! This is interesting from a factual point of view, as the extended formulation contains no new factual information. A pure propositional reasoner would still have come to the same answer independent of the truth value of the question, as the additional information about what "most people say" does not factually contribute to comparing two different weights. It is rather a hint on the meta level that is often used in human communication to indicate the presence of a pitfall or trick question, and therefore not to choose the 'obvious' answer. In the presence of such a hint GPT-3 does not just negate its previous given answer but arrives at the fact that both weigh the same. Seeing LLM considering these meta level hints might help with future algorithms that integrate LLMs into their processes. Providing textual hints might be comparable to switching between different heuristics as it is done in today's classical algorithms.

**Chain of Thoughts Prompting.** In contrast to humans, current LLMs are strongly dependent on specific prompt engineering for picking up task descriptions. Several methods, such as Chain of Thoughts (CoT, Wei et al. (2022)) or Tree of Thoughts (Yao et al., 2023) condition the model on an answering style that promotes solving the given task by tackling its individual substeps. Thus, often resulting in improved consistency and

---

[17]Except for GPT-4. Please consult the 'Newer General-Purpose LLMs.' paragraph below

| | Rolling (8) | Support (8) | Intuitive Physics Collisions (4) | Seesaw (4) | Weights (5) | Tools (7) | Accuracy |
|---|---|---|---|---|---|---|---|
| GPT-3 | **6** | **5** | **4** | **2** | **2** | 3 | **61.11%** |
| Luminous | 1 | 0 | 0 | 1 | 1 | 2 | 11.11% |
| OPT | 2 | 0 | 1 | 0 | 0 | **4** | 19.44% |
| GPT-4 | **7** | **8** | **4** | **3** | **5** | **5** | **91.66%** (!) |

| | Causal Chains (Basic Prop. Logic) N=2 | 3 | 4 | 5 | 6 | 7 | 8 | 9 | 10 | Subchains (4) | Randomized (7) | Accuracy |
|---|---|---|---|---|---|---|---|---|---|---|---|---|
| GPT-3 | | ✓ | ✓ | ✓ | | | ✓ | | ✓ | 2 | 2 | 45.00% |
| Luminous | ✓ | | | | ✓ | ✓ | ✓ | ✓ | | 1 | 4 | 50.00% |
| OPT | | ✓ | | | ✓ | | | | | 0 | 2 | 20.00% |
| GPT-3 (CoT 4,6) | ✓ | ✓ | ✓ | ✓ | ✓ | ✓ | ✓ | ✓ | ✓ | 4 | 7 | **100.00%** |
| Luminous (CoT 1) | ✓ | ✓ | ✓ | ✓ | ✓ | ✓ | ✓ | ✓ | ✓ | 3 | 3 | 75.00% * |
| OPT (CoT 4) | ✓ | ✓ | ✓ | ✓ | ✓ | ✓ | ✓ | ✓ | ✓ | 3 | 4 | 80.00% * |
| GPT-4 | ✓ | ✓ | ✓ | ✓ | ✓ | ✓ | ✓ | ✓ | ✓ | 4 | 7 | **100.00%** (!) |

| | Natural Word Chain Real World (5) | Imaginary (6) | Mixed (4) | Accuracy |
|---|---|---|---|---|
| GPT-4 | 4 | **6** | 3 | 86.66% |
| GPT-3 | 3 | 0 | 2 | 33.33% |
| Luminous | 2 | 3 | 2 | 46.66% |
| OPT | 2 | 0 | 2 | 26.66% |
| GPT-4 (CoT 3,4) | **5** | **6** | **4** | **100.00%** |
| GPT-3 (CoT 2) | **5** | 3 | 3 | 73.33% |
| Luminous (CoT 4) | 2 | 5 | 2 | 60.00% |
| OPT (CoT 1,4) | 3, 1 | 5,**6** | 2,3 | 66.66% |

Table 1: (top) Querying LLMs for questions about intuitive physics. GPT-3 performs reasonably well across all queried areas, while OPT performs better on the Tools subtask. (bottom) Querying LLMs for propositional logic on causal chains. GPT-3 and Luminous perform equally on causal chains of varying length. Luminous outperforms the other models when asked about causal chains with randomized chain ordering or variable names. Overall with plain prompts LLMs display a mixed performance. Results with Chain of Thoughts prompting (CoT) yield improved results as the model is able to correctly carry out required substeps. We report the best performing CoT model. The number of provided examples is noted besides the model names. All results are shown in the Appendix. Improvement of GPT-4 with CoT is mostly due to better adherence to the expected answer style by excluding discussion of alternative answers and answering "Yes" or "No" more decisively. The additional CoT examples strongly bias Luminous and OPT models into answering all queries with either only "Yes" (or "No") for certain CoT settings. These results are marked with (*). GPT-4 results of the 'Intuitive Physics' and 'Causal Chains' data sets marked with (!) may have been part of the GPT-4 training data. Differences in prediction accuracy with natural variable names instead of symbols were probed with the Natural Word Chain data set. No significant difference to the symbolic setting was observed.

overall accuracy. In this paper we carried out additional experiments with Chain of Thoughts prompting on the Causal Chains and Natural Word Chain dataset. For CoT prompting a series of question answer-pairs that match the style of the actual question are provided to the model. We provide up to eight question-answer pairs of the following style: "Q: If Y causes Z and X causes Y. Does X cause Z? <linebreak> A: Because X causes Y and Y causes Z, X causes Z. The answer is yes.", with mixed, positive and negative, examples. The provided QA samples and all CoT results can be found in Appendix C.5. The provided examples contain the same style, but use an exclusively distinct set of variable names compared to the actual questions. Table 1 shows strong improvements for all models on both data sets with CoT querying.

Humans utilise similar kinds of thought processes to solve complex tasks. However, the explicit provision of examples has a three-fold implication on our evaluation. First, it provides the model with the expected answer format. This effect is clearly seen, as the answers of all models adapt the answering style from the previously provided examples. The previously worst model, OPT, is now able to output 'Yes' or 'No' answers,

where is previously often only repeated the input prompt, resulting in a wrong answer. Secondly, CoT seems to induce a strong bias to certain kinds of models. Specifically Luminous and OPT tend to only answer 'Yes' or 'No' for specific CoT setups. While this greatly improves performance–always answering 'Yes' leads to a 75% accuracy on Causal Chains–it it clear that these models do not improve in understanding of the problem through the chain of thought. Third, this kind of method requires knowledge about the expected type of answer. More specifically, the models are provided with an exemplary 'thought process' to solve a given task. By providing these 'chains of thoughts' we 'materialise' the otherwise unobserved thought process and provide it via the means of input data. As LLMs are few-shot learners (Brown et al., 2020), we are no longer able to distinguish between the models' understanding of the task and the model simply mirroring the provided thought process. While such behaviour can be interpreted as a flaw of the models under consideration, we want to point out that humans sometimes adopt similar behaviour by carrying out a certain sequence of actions to solve a task. For example, children may solve certain maths problems by applying steps of an algorithm rather 'mechanically' than understanding the actual implications of these steps towards producing the right answer. Real understanding of the problem, however, can only be tested by the models solving a problem via induction without the provision of exemplary solution paths. This setup would require the models to only take the description of a task and utilize the given information towards solving it. As no examples are allowed to be provided, this rules out CoT style querying and in turn decreases performance.[18]

**Newer General-Purpose LLMs.** We've further run our evaluation on GPT-4 (OpenAI, 2023), which poses the successor to GPT-3 and is the only model in the evaluation that was trained at a time where a prior version of this work (with a subset of the presented experiments in place) was readily available on platforms such as arXiv and other conference proceedings. In summary, GPT-4 performs better in the "common sense" category, whereas performing just "as bad" in the following "specific ground truth" category of the experiments. This is in the realm of our prior expectations, however, it is worthwhile to note that in the former category GPT-4 performing well might in fact be due to our data being part of GPT-4's training data, since (as mentioned before) a prior workshop paper version of this work was already made publicly available on arXiv and other conference proceedings, but furthermore by us performing experiments via the API we actually officially agree with the companies terms of being allowed to use our queries, thus being available at the time of GPT-4 training (unlike GPT-3 and all other models like OPT and Luminous previously evaluated). Unfortunately, we do not have a way of proving this conjecture. Regarding the CCF conjecture and further the prior discussion on meta SCM and their relation to fine-tuning, we could consider GPT-4 as a fine-tuned version of GPT-3, where our data was part of the training data. We note again that this is a purely speculative comment based on all established results.

## 4.2 Methods and Results for Causal Discovery on Ground Truth

Previously, we have seen that LLMs do not necessarily perform well on tasks where they are not just required to know a fact but also to *reason* with that fact. For the following experiment, the LLM will not be required to reason with facts but it will be sufficient to simply recall the right facts. Therefore, we can now simply resort to causal discovery benchmarks for evaluating whether LLMs can recover causal graphs when queried accordingly. We propose a naïve causal discovery approach for LLMs to this end. Fig.2 provides a schematic overview of the naïve structure discovery procedure that we propose. To account for stability and reproducibility, we present different wordings to the queries (synonymous formulations) and disable parameters that

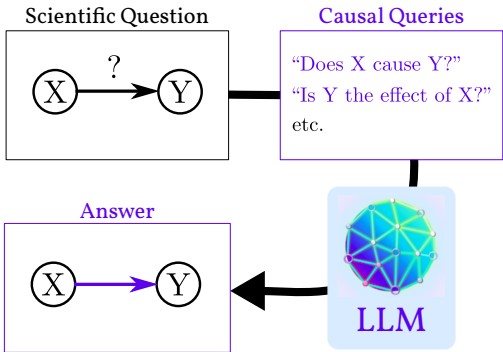

Figure 2: **Naïve Causal Discovery with LLMs.**

---

[18]It could be argued that LLMs might be trained to reason inductively about arbitrary problems. While this provides an interesting future research direction, the analysis is out of scope for this paper.

| | Metric | Altitude | Health | Driving | Recovery | Cancer | Earthquake | LLM |
|---|---|---|---|---|---|---|---|---|
| *Causal Graph* | SID ↓ | $\mathbf{0.80}_{\pm 0.40}$ | $\mathbf{7.20}_{\pm 0.75}$ | $3.00_{\pm 0.89}$ | $\mathbf{4.00}_{\pm 1.79}$ | $11.80_{\pm 4.66}$ | $\mathbf{11.40}_{\pm 1.50}$ | GPT-3 |
| | | $1.40_{\pm 0.80}$ | $9.80_{\pm 2.99}$ | $\mathbf{2.40}_{\pm 1.20}$ | $4.00_{\pm 2.53}$ | $13.20_{\pm 7.55}$ | - | GPT-4 |
| | | $1.20_{\pm 0.98}$ | $10.60_{\pm 1.85}$ | $6.00_{\pm 0.00}$ | $5.40_{\pm 1.20}$ | $\mathbf{11.40}_{\pm 3.07}$ | $16.00_{\pm 3.63}$ | Luminous |
| | | $1.60_{\pm 0.80}$ | $10.80_{\pm 2.40}$ | $5.00_{\pm 1.26}$ | $5.80_{\pm 0.40}$ | $16.80_{\pm 1.94}$ | $15.60_{\pm 5.95}$ | OPT |
| | SHD ↓ | $0.80_{\pm 0.40}$ | $\mathbf{4.00}_{\pm 0.63}$ | $2.60_{\pm 0.49}$ | $\mathbf{2.20}_{\pm 0.40}$ | $\mathbf{7.00}_{\pm 1.41}$ | $\mathbf{4.60}_{\pm 0.80}$ | GPT-3 |
| | | $0.80_{\pm 0.40}$ | $6.20_{\pm 2.23}$ | $\mathbf{1.60}_{\pm 0.80}$ | $2.80_{\pm 1.60}$ | $7.40_{\pm 1.62}$ | - | GPT-4 |
| | | $\mathbf{0.60}_{\pm 0.49}$ | $7.00_{\pm 1.10}$ | $4.20_{\pm 0.40}$ | $3.40_{\pm 0.80}$ | $10.00_{\pm 3.52}$ | $5.60_{\pm 1.62}$ | Luminous |
| | | $0.80_{\pm 0.40}$ | $7.40_{\pm 1.20}$ | $3.40_{\pm 1.20}$ | $4.00_{\pm 0.00}$ | $13.20_{\pm 1.60}$ | $8.60_{\pm 3.01}$ | OPT |
| *ML* | $F_1$ Score ↑ | $0.20_{\pm 0.40}$ | $0.47_{\pm 0.14}$ | $0.11_{\pm 0.23}$ | $0.27_{\pm 0.33}$ | $0.35_{\pm 0.11}$ | $\mathbf{0.12}_{\pm 0.15}$ | GPT-3 |
| | | $0.60_{\pm 0.33}$ | $\mathbf{0.55}_{\pm 0.06}$ | $\mathbf{0.64}_{\pm 0.10}$ | $\mathbf{0.63}_{\pm 0.19}$ | $\mathbf{0.51}_{\pm 0.04}$ | - | GPT-4 |
| | | $\mathbf{0.80}_{\pm 0.16}$ | $0.41_{\pm 0.21}$ | $0.46_{\pm 0.09}$ | $0.55_{\pm 0.07}$ | $0.40_{\pm 0.13}$ | $0.40_{\pm 0.04}$ | Luminous |
| | | $0.73_{\pm 0.13}$ | $0.52_{\pm 0.05}$ | $0.53_{\pm 0.15}$ | $0.47_{\pm 0.07}$ | $0.35_{\pm 0.03}$ | $0.47_{\pm 0.07}$ | OPT |
| *Edges* | Sparsity | $0.90_{\pm 0.20}$ | $0.63_{\pm 0.28}$ | $0.77_{\pm 0.31}$ | $0.70_{\pm 0.31}$ | $0.65_{\pm 0.16}$ | $0.93_{\pm 0.07}$ | GPT-3 |
| | | $0.30_{\pm 0.40}$ | $0.22_{\pm 0.31}$ | $0.60_{\pm 0.25}$ | $0.20_{\pm 0.27}$ | $0.45_{\pm 0.11}$ | - | GPT-4 |
| | | $0.20_{\pm 0.24}$ | $0.22_{\pm 0.35}$ | $0.03_{\pm 0.07}$ | $0.10_{\pm 0.13}$ | $0.40_{\pm 0.16}$ | $0.74_{\pm 0.12}$ | Luminous |
| | | $0.10_{\pm 0.20}$ | $0.05_{\pm 0.10}$ | $0.17_{\pm 0.21}$ | $0.07_{\pm 0.13}$ | $0.18_{\pm 0.12}$ | $0.41_{\pm 0.18}$ | OPT |
| | ADS ↑ | $0.50$ | $\mathbf{0.62}$ | $\mathbf{0.33}$ | $0.50$ | $0.69$ | $0.00$ | GPT-3 |
| | | $0.50$ | $0.61$ | $0.17$ | $\mathbf{0.83}$ | $\mathbf{0.85}$ | - | GPT-4 |
| | | $\mathbf{1.00}$ | $0.53$ | $0.17$ | $0.17$ | $0.38$ | $0.26$ | Luminous |
| | | $0.50$ | $0.25$ | $0.25$ | $0.33$ | $0.28$ | $\mathbf{0.47}$ | OPT |

Table 2: Comparing LLMs prediction to existing ground truth causal structures. The metrics concerned with the causal graph structure (SID, SHD) reveal a closer match of GPT-3 and GPT-4 predictions to the ground truth causal structures than for the other LLMs. High $F_1$ Scores and low sparsity indicate densely connected graph prediction by Luminous and OPT. This can be desired for ML applications. The ADS reveals that all LLMs increase their decisiveness on edge directions when querying with asymmetric sentence templates. Metrics for GPT-4 on Earthquake are not computed to prevent skewed results due to unclear judgement of meta answers.

induce randomness (e.g. temperature), respectively. It is important to note that the proposed naïve structure discovery procedure is not a proper induction method in the classical sense as it does not use actual data as input to perform the inferences (all the possible inferences are established upon training completion). In that sense, LLMs behave much like humans, who simply recall that "a higher altitude means a lower temperature" than to look at actual data recordings of altitude and temperature (and other variables) to perform the causal inference. As anticipated, the LLM thereby also inherits natural language ambiguities. To give an example, even if the LLM is prompted with an additional "Answer with Yes or No" the LLM is not constrained to oblige. We use five different query wordings (or formulations) such as "Are $X$ and $Y$ causally related?" or "Does $X$ cause $Y$?" (see appendix for full list). The first three of which are classified as *symmetric* queries since the expected answer is a mere association $X$–$Y$ and the last two wordings classify as *asymmetric* accordingly i.e., we expect either $X \rightarrow Y$ or $X \leftarrow Y$ (in the case of an existing relation). For any answer given by the LLMs we automatically classify answers starting with 'Yes' or 'No' accordingly and manually label the remaining ones. For any data set containing $N$ variables there are $\binom{N}{2}$ possible edges. Since the edges are directed we get twice the amount of queries, one query for each direction, multiplied by the number of query wordings $Q = 5$ and arrive at $2 * \binom{N}{2} * Q$ queries per data set. We consider publicly available data sets that propose a 'ground truth' causal graph (which depicts the data generating process). We consider six data sets: altitude (A; Mooij et al. (2016)), health (H; Zečević et al. (2021)), recovery (R; Charig et al. (1986)), driving (D; synthetic), cancer (C) and earthquake (E), both (Korb & Nicholson, 2010). For all data sets we query the LLM with all possible combinations of edges between any two variables. For our data sets, we get 10 questions for Altitude, 100 for Cancer, 60 for Health, 30 for Driving, 100 for Earthquake and 30 for Recovery respectively.

After querying all edges for all data sets we then compare each of the obtained graphs to the respective ground truth using different metrics. In total we discuss two key observations. Table 2 presents

the results of our experiment. We present metrics to measure different aspects of the LLM predictions. We measure fitness to the *causal graph* using Structural Intervention Distance (SID; Peters & Bühlmann (2015)) and Structural Hamming Distance (SHD; Acid & de Campos (2003); Tsamardinos et al. (2006)). *Machine Learning* (*ML*) applications might care more about capturing all relevant connections and less about including too many. To reflect this we report the $F_1$ score. Furthermore, we inspect individual statistics on the *edges* of the predicted graph when wording of the queries changes.

We define sparsity as the number of predicted edges in relation to the maximum number of possible edges.[19] Intuitively, this metric measures the percentage of maximally predictable edges and gives an insight on whether a LLM tends to predict sparse ($Sparsity \approx 1$) or densely connected graphs ($Sparsity \approx 0$). Additionally we want to measure the amount of directed edges within a graph. While SCM are not restricted to DAG structures, most classical causal data sets are modeled using DAGs. We further introduce a novel metric based on the model's *decisiveness d* which is defined as the percentage of directed edges within the total number of predicted edges. The exact procedure is shown in Algorithm 1 in the appendix. When comparing two graph structures we define $\Delta d$ to be the change in decisiveness between two predictions: $\Delta d(A, B) := d(B) - d(A)$. A positive $\Delta d$ value indicates an increase in directed edges when switching from A to B and vice versa. In particular we can now define our novel metric called *asymmetric decision score* (*ADS*) as $\Delta d$(Symmetric, Asymmetric) to measure the change in decisiveness when switching from symmetric to asymmetric sentence wording (averaged over all sentence templates).

**Earthquake**

Legend: [E]arthquake [B]urglary [A]larm [J]ohn calls [M]arry calls

Figure 3: **Meta answers for unknown concepts.**

**Meta-Answers** Additional experiments where conducted for GPT-4. In contrast to the older LLMs, GPT-4 gives meta answers for the Earthquake data set. Specifically, GPT-4 acknowledges that the variables 'John calls' and 'Marry calls' refer to specific persons for which no further information is available. As such, no statement can be made about whether John or Marry would call in the event of an earthquake, burglary or alarm going off. As shown exemplarily for the 'related' template in Fig. 3, either no connection is predicted or a meta answer (purple) is given for all queries involving 'John' or 'Marry'. This pattern holds consistently across all query templates. For computing metrics, meta answers could be filled by an oracle with access to the specific correct information, e.g. the correct edge orientation of the ground truth SCM. However, a model could cheat by stating 'insufficient information' for as many edges as possible to leverage ground truth information and increase scores. Due to the unclear handling of meta answers we choose to not compute metrics for GPT-4 on the Earthquake data set.

Results are shown in Table 2. For (almost) all six data sets we observe a better compliance to the causal graph structure for GPT-3 than for the other models. Looking at the sparsity, we observe that GPT-3 predicts much sparser graphs in relation to Luminous and OPT. This mode of sparse predictions matches well with the ground truth graphs, which are generally also sparsely connected. In cases where the exact causal structure is relevant we would prefer GPT-3 over the others. In return Luminous and OPT feature better $F_1$ scores, as they predict more edges to be present. This might be favourable for ML applications where false negative predictions are lowering performance and we only want to prune out clearly non existing edges. Overall, predictions of the causal structure and for individual edges are noisy. Depending on the use case GPT-3 or Luminous and OPT might be better suited. Consider ADS

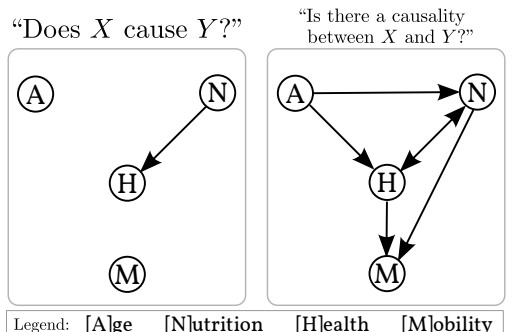

Legend: [A]ge [N]utrition [H]ealth [M]obility

Figure 4: **Sensitivity to Query Wording.**

in Table 2. As discussed before ADS is positive when more edges are predicted as directed when using one of the asymmetric sentence wording than when using a symmetric one. We observe that *without exception*

---

[19]The technical definition can be found in the appendix.

| | Metric | Altitude | Health | Driving | Recovery | Cancer | Earthquake | Method |
|---|---|---|---|---|---|---|---|---|
| *ML Caus. Graph* | SID ↓ | $\mathbf{0.80}_{\pm 0.40}$ | $\mathbf{7.20}_{\pm 0.75}$ | $3.00_{\pm 0.89}$ | $4.00_{\pm 1.79}$ | $11.80_{\pm 4.66}$ | $\mathbf{11.40}_{\pm 1.50}$ | Direct |
| | | $1.00_{\pm 0.00}$ | $9.20_{\pm 3.12}$ | $\mathbf{1.60}_{\pm 1.36}$ | $\mathbf{3.60}_{\pm 1.36}$ | $\mathbf{10.40}_{\pm 2.06}$ | $16.00_{\pm 3.29}$ | k-NN |
| | SHD ↓ | $\mathbf{0.80}_{\pm 0.40}$ | $\mathbf{4.00}_{\pm 0.63}$ | $2.60_{\pm 0.49}$ | $\mathbf{2.20}_{\pm 0.40}$ | $7.00_{\pm 1.41}$ | $\mathbf{4.60}_{\pm 0.80}$ | Direct |
| | | $1.00_{\pm 0.00}$ | $6.80_{\pm 2.32}$ | $\mathbf{2.00}_{\pm 1.41}$ | $3.20_{\pm 1.17}$ | $\mathbf{5.80}_{\pm 1.33}$ | $11.40_{\pm 1.50}$ | k-NN |
| | $F_1$ Score ↑ | $\mathbf{0.20}_{\pm 0.40}$ | $\mathbf{0.47}_{\pm 0.14}$ | $0.11_{\pm 0.23}$ | $0.27_{\pm 0.33}$ | $0.35_{\pm 0.11}$ | $0.12_{\pm 0.15}$ | Direct |
| | | $0.00_{\pm 0.00}$ | $0.28_{\pm 0.19}$ | $\mathbf{0.61}_{\pm 0.34}$ | $\mathbf{0.41}_{\pm 0.26}$ | $\mathbf{0.46}_{\pm 0.06}$ | $\mathbf{0.20}_{\pm 0.13}$ | k-NN |
| *Edges* | Sparsity | $0.90_{\pm 0.20}$ | $0.63_{\pm 0.28}$ | $0.77_{\pm 0.31}$ | $0.70_{\pm 0.31}$ | $0.65_{\pm 0.16}$ | $0.93_{\pm 0.07}$ | Direct |
| | | $1.00_{\pm 0.00}$ | $0.57_{\pm 0.12}$ | $0.47_{\pm 0.19}$ | $0.40_{\pm 0.23}$ | $0.67_{\pm 0.09}$ | $0.47_{\pm 0.15}$ | k-NN |
| | ADS ↑ | $\mathbf{0.50}$ | $\mathbf{0.62}$ | $\mathbf{0.33}$ | $\mathbf{0.50}$ | $\mathbf{0.69}$ | $\mathbf{0.00}$ | Direct |
| | | $0.00$ | $0.48$ | $-0.56$ | $0.08$ | $-0.03$ | $-0.03$ | k-NN |

Table 3: Results for predicting causal structures with existing ground truth graphs. We compare direct predictions of Table 2 (direct) to embedding prediction of the GPT-3 Ada model ('text-embedding-ada-002') with using nearest neighbours (k-NN). Predictions with k-NN perform comparable to direct querying, improving on the Driving, Recovery and Cancer data sets for causal graph and ML metrics. However, positive ADS values vanish for k-NN in comparison to directly querying LLMs, implying that k-NN does not respect asymmetric query wording.

all LLMs increase their decisiveness when queried with an asymmetric wording. While this observation is consistent with the natural interpretation that an asymmetric query like "Does $X$ cause $Y$?" only accepts the answers $X \to Y$ and $X \quad Y$, but not $X \leftrightarrow Y$, the observation is still surprising as there are no formal guarantees to the query that this should be the case. It might suggest that the LLM indeed learned the difference between the two types of questions on a causal level. While Luminous and OPT remain overall stable in their prediction across data sets and wordings, GPT-3 reacts with *unsmooth change to alternate wordings.* Consider Fig.4 where a significant change in the predicted graph is observed simply by changing the query wording. A possible interpretation for this observation is that a keyword such as 'causality' might be embedded further away from an alternate keyword (here for instance 'cause') within the LLM's latent space, thus answering correctly.

Reflecting upon the observed results in our experiments that show lacking of LLMs, especially in consideration of otherwise positive results in the literature regarding the causal inference capabilities of general-purpose LLMs (like for instance in (Kıcıman et al., 2023)), we add the following comment: (i) the results we present do not contradict prior literature but rather complement further understanding of LLM capabilities. Taking (Kıcıman et al., 2023) into account, the authors present a discussion of a prior iteration of this present work, placing it into context with their own results. (ii) observing a high accuracy in the Tübingen cause-effect pairs data set by Mooij et al. (2016) (especially when not conditioning on the actual data) does not imply LLMs being capable of causal inference. Put with the words 'understanding' and 'knowing,' it is not implied that they understand causality and in fact, it does not even imply LLMs knowing causality. While the observed LLM performance reported could be considered impressive at first glance, looking at the actual data set reverses the conclusion. The data set can be found at: https://webdav.tuebingen.mpg.de/cause-effect/. To first quote Mooij et al. (2016): "[W]e do not guarantee that all provided ground truths are correct." The data set consists of 108 $(X, Y)$ pairs for which we don't necessarily have ground truth (whether $X \to Y$ or vice versa). Like in our setup, the actual data measurements are not used, but only the concepts like for `pair0001` which is $X$: `altitude` and $Y$: `temperature`. This very same example was employed in our experiments as well and is arguably reasonable since $X, Y$ are 'common sense'/every-day concepts. But what about example `pair0085` for instance where $X$: `time to take weekly measurements (from 1 to 14)` and $Y$: `protein content of the milk produced by each cow at time X`? While not cherry-picked, this example illustrates how 'good' LLM prediction results (measured simply in terms of "hit-and-miss" accuracy) are meaningless. In conclusion, both (i) and (ii) highlight that the overall discussion is nuanced and that prior literature is in favor of our paper's results

### 4.3 Method and Results for Knowledge Base Fact Embeddings

The reason that we observe LLMs to perform with mixed results might lie in the simple fact that LLMs have not (and are not capable of) memorizing all the causal facts available in training. We therefore create an artificial causal 'signal' by using existing causal information from a knowledge base and taking LLM embeddings of its facts. However, it is unclear to which extent LLM embeddings encode knowledge e.g. whether it is simply a 'code' memorization or some higher order features such as meaning. Depending on the strength of the embedding model, text embeddings turn out to be nothing more than a symbolic representation of the embedded text, containing a one-to-one representation of the natural language words. While these simple-most embeddings might be easy to generate, they are not well suited for semantic similarity search. Every change in the text will offset the following symbols, resulting in a low embedding similarity between two similar texts. A much better approach is the encoding of knowledge base facts and their relations to each other into the embedding vector. This removes the dependence on a word-by-word encoding procedure and allows us to encode embedding similarity on a conceptual level. While LLMs might encode all sorts of information, we expect that also causal information will be encoded in this way. In an ideal case the embeddings should contain information about the talked-about concepts of cause and effect, while additionally encoding the direction of the causal edge. ConceptNet (Speer et al., 2017) is a knowledge graph combining multiple data sources, thus containing a large range of relational information. Among other things, the ConceptNet contains explicit information about causal connections ("/r/Causes/" relations). While the strength of the information varies among the different entries, we get hold of a causal signal for real world relations. Filtering ConceptNet for causal edges results in 1,282 unique causes expanding to 16,567 individual cause-effect pairs. For every causal edge we generate text embeddings using the GPT-3 Ada Model (text-embedding-ada-002). We generate the statement sentences to be embedded using our 5 sentence templates (e.g. Rain → Floods is instantiated as "Rain causes floods.", "Rain and floods are causally related.", ...) and additionally generate the same amount of anti-causal samples by swapping the cause and effect of the edges. In total, we get hold of 165,670 causal and anti-causal embeddings. In the following, we will again be evaluating the performance against ground truth causal graphs using the previous metrics but with the change that we don't query the LLM directly but rather do a matching based on projections of the knowledge base facts. Since we have access to the ConceptNet edges, we can confirm that relevant causal knowledge for our prediction tasks is indeed contained in the set of embeddings. To give a particular example, we find for instance that a relation of the Driving data set, driving style → remaining fuel, is present in the ConceptNet data. The expressions used in ConceptNet and our query do not match in wording (see Fig. 5), but should be *semantically* similar enough to serve as a causal signal for LLM prediction. For doing the evaluation, we run a nearest neighbour prediction (k-NN with k=1) with cosine similarity. Like in the previous experiments we build causal graphs to determine whether or not there is an edge between every possible combination of two variables of the data set. We compare the undecided edge embedding of the data set to all statements of the ConceptNet data with known labels. The presence of an edge is decided based on the label with the most similar embedding. That is, we decide for the edge to be present if the most similar embedding stems from a causal fact of the ConceptNet-based data set. If the nearest neighbour stems from an anti-causal fact, we predict the edge to be absent.

We discuss our findings on the results shown in Fig. 5. We find that a robust edge prediction emerges for the previously mentioned driving style → remaining fuel edge of the Driving data set. Looking at the nearest neighbours that are used for deciding the presence of the edge, we find that all templates match to the driving → lack of fuel ConceptNet fact as their nearest neighbor, with an exception for the 'Influence' template which matches to the Moving car → use fuel fact. In the case of the 'Cause' template we find that the nearest neighbour erroneously is matched to the anti-causal lack of fuel → driving fact, and in turn is predicted non-present. For the Cancer data set we even observe a word-by-word correspondence of Smoking → Cancer which results in a perfect prediction of the edge for all five sentence templates (see Appendix). For the other data sets, for which we could not confirm related facts in ConceptNet, the same unreliable results like in the direct querying experiment are observed. In Tab. 3 we compare the k-NN results to the direct querying of GPT-3 from our previous experiment. We find improvements on tree data sets, including Driving and Cancer for which we confirmed facts in ConceptNet. For Driving and Cancer we simultaneously observe an increase in $F_1$ score indicating, that the improvements come from an actual optimized graph structure and not only as a result of predicting a sparser graph. A downside of k-NN

Figure 5: **Transfer of ConceptNet Causal Knowledge into Graph Predictions.** Facts about driving influencing the fuel consumption can be found in the ConceptNet data (top). As a result the related edge "[D]riving style → [F]uel consumption" of the driving data set gets predicted correctly in 4 out of 5 sentence wordings when applying k-NN classification. All templates match to the driving → lack of fuel ConceptNet fact as their nearest neighbor, except for the "Influence" template which matches to Moving car → use fuel.

prediction is the emergence of negative ADS scores in three of the six data sets. Indicating, that the LM might not encode asymmetric aspects of query texts into embeddings.

## 5 Related Work

This present work takes inspiration from various recent results. Yet, to the best of our knowledge, it is the first to investigate the question in its presented form, especially in terms of formalization and overarching hypothesis that serves as a candidate *explanation* for conclusions we make from LLM input-output behavior. For instance, Wang et al. (2021) leveraged BERT as the underlying foundation model to perform inferences according to the rules of Pearl's *do*-calculus (Pearl, 2009). This allows for causal inference with the foundation model as an 'inference engine', but it misses out on the question of how causal the models themselves might be to begin with. Another work, by Khetan et al. (2022), is closer to our work in the sense that causal relations are queried by natural language directly, however, the subject of interest is orthogonal to both the ongoing debate and the investigation presented in this work. On another note, McMilin (2022) investigated selection bias within LLMs by first arguing about reasonable causal modelling assumptions and then validating them empirically. Discarding causality but with the arguably identical goal of understanding what LLMs are capable of, Zhang et al. (2022a) investigated an approach using propositional logic that concluded that LLMs only learn statistical features that inherently exist in logical reasoning problems. Also noteworthy are works such as conducted by Talmor et al. (2022) where the goal is to create a benchmark that makes explicit the deficiencies (if existent) of LLMs, which can be understood as a complementary goal to understanding how the models work in the first place. Lastly we want to refer to other approaches that extract (causal) question-answer pairs from text sources (Ho et al., 2023; Bondarenko et al., 2022). While both of these data sets might provide a causal ground truth, they do not compose further graph structures out of the extracted data and serve mainly as evaluation metric for LLM performance. Therefore, there is no distinction between 'understanding' and 'knowing', however, with the benefit of being useful to *improving* future LLMs since we can evaluate their ranking. As a final note on related work, Kıcıman et al. (2023) recently discussed the impressive feats of LLMs on the Tübingen pairs data set (Mooij et al., 2016) for which we comment (a) that the data set itself is not provided with any form of 'absolute' ground truth in that some of the pairs are in fact "educated guesses," and (b) the observations are still consistent with the proposed theory and observed empirical results in this work. Furthermore, the authors acknowledge in their penultimate section a prior version of this present work, which described "correlations of causal facts" on an intuitive level, as a reasonable explanation for their observed phenomena.

Putting the present work more broadly into the context of works on the intersection of causality and natural language processing: Abraham et al. (2022) interpret interventions as text manipulation on semantic concepts where the implicit assumption lies in the exogenous terms of an SCM being the 'style' of the text (e.g. synonyms and how a given content is being phrased in a sentence). They present a data set on which

they can measure various causal quantities. The idea is to benchmark to which extent NLP models respect 'realistic' semantic interventions. Another benchmark was proposed by Jin et al. (2023) that, like our work, also tackles causation and was made public shortly after to our work, where the authors intend on performing causal discovery with both off-the-shelf and fine-tuned LLMs. The authors came to a similar conclusion as our work on the empirical part since the off-the-shelf LLMs perform poorly, whereas fine-tuned ones seemed promising. This is further consistent with our conjecture on GPT-4 being a "fine-tuned GPT-3," when it comes to the presented causality experiments. In a survey paper, Feder et al. (2022) more broadly discussed how causality can actually benefit NLP tasks (but also how causal effects could be estimated), which is mostly orthogonal to the question discussed in this present work but shares the commonality of explicating assumption in current models like LLMs.

## 6 Conclusive Discussion

We have multiple reasons to believe that LLMs are not causal (i) them obviously being only trained on textual data not physical measurements (see Fig.1) which prohibits any sort of induction on the actual data-generating mechanism, (ii) them not having any causal assumptions marked out explicitly (as for instance having explicitly modelled structural equations like neural causal models), and (iii) the Causal Hierarchy Theorem prohibits any causal inference from purely observational data for any model, thus including LLM. However, the fact that we do observe LLMs perform well *occasionally* on causal inference tasks, as our empirical part of the analysis has shown, stands in stark contrast to (i-iii) which would justify a statement of the form that LLMs are only "castles in the air" as seen in the very beginning in the motivation to our present paper. Fortunately, our key theoretical contribution, the definition of meta SCM (Def.3) and the correlations of causal facts conjecture (Conj.1), have provided a *sound explanation* for the apparent contradiction (or even paradoxon). The following two paragraphs give another summarizing account of both theory and empirics in this paper, respectively. A more complete summary of the main idea of the paper can be found in Sec.2.

As we started exploring in Sec.3, physical reality (or nature) ultimately dictates any sort of *causal assumptions* that we can end up formalizing in that it is the 'ultimate' data generating process to which we attach truth values. That is, the graph that captures the idea of the textual statement "altitude causes temperature" but also of a related textual question like "Does altitude cause temperature?". Obviously, changing the description of either through intervening on one would not change the other, giving us reason to believe that there is no direct causation between them. Still, they are clearly confounded. In our physical reality, the given textual statement on altitude and temperature corresponds to the truth, therefore, being factually correct. We can expect such a fact to be encoded not just in encyclopedia articles like on Wikipedia but more widely spread across sources, all of which ultimately play a part in the LLMs vast training data. We can further expect a correlation between these factual statements and corresponding questions. We conjecture that the LLMs, in the case where they behave correctly when queried causally, have learnt to exploit these correlations. While at the core lies this high level idea, we were fortunately able to formalize it consistently with the theory of causation by Pearl. In classical causality literature, our data usually expresses low-level (physical) quantities and what makes the model causal are actually the causal assumptions. However, there is no restriction on what the variables might denote. We might have a 'big' SCM (that might be considered as *nature itself* which is an idea that we can also link to the concept of a *Universal* Turing Machine (Turing et al., 1936)) which generates other SCMs so to say i.e., the data talks about causal assumptions. In other words, this 'meta' SCM generates, as observations, abstractions of causal quantities.

Then in Sec.4 we conducted an empirical analysis in search of evidence to the CCF conjecture. We identified three components to proving the conjecture (i) identify the proposed meta SCM, (ii) show that using causal facts as answers is optimal and (iii) show that LLMs indeed give the causal facts as answers to the causal question. Unfortunately, (i-ii) seem generally infeasible which is why we focussed on studying (iii). To this end, we measured LLM performance systematically (a) in "common sense" settings like reasoning and intuitive physics, (b) in settings where the causal graph is (partially) known and (c) when using their embeddings of knowledge base facts. While we do not intend on summarizing all of Sec.4, we will take a big picture perspective on our answer for (iii). We believe that (iii) does hold since in the cases where the LLMs answer causal questions correctly since we found evidence that they indeed uses causal facts that

could be expected to be found in the training data. With that being said, we see it as favoring evidence to our CCF conjecture, however, a definitive answer cannot be given because for one, (i-ii) are yet to be proven and secondly, the LLMs underperform way too often. That is, they might be "causal parrots" but rather underwhelming ones in that they do not recite everything that one would want them to.

### 6.1   Takeaway, Ethical Challenges and Societal Implications.

We believe there to be two take-away messages of this initial effort to resolving the mysteries around (causal) reasoning capabilities of LLMs. To start with the negative message, it is to say that we can *not* rely solely on LLMs as we cannot expect any sort of generalization in terms of causal prowess. Current LLMs are unable to process actual physical data measurements to ground their available textual facts. This prohibits LLMs from doing actual, inductive inference like classical (causal) structure discovery methods for instance do. However, the positive message to the story is that we can use the LLMs as a head start to learning and inference. In that sense, they might very well serve as stepping stones towards progress in AI/ML research. On the ethical side and societal scale, it ultimately also inherits all concerns revolving around AGI itself. While in our work we did not encounter any noteworthy ethical challenges such as for instance racial bias, as can easily happen when working with LLMs, we did uncover bias in its views on e.g. medical topics as illustrated by the results on the medical causal graph. However, in that sense, any predictive model has a certain bias, it is just less apparent for LLMs. Importantly, we do want to raise one crucial societal discussion point around *learning from facts.* Arguably, the ideal should be *understanding* and not just *knowing*, since the latter lacks both generalization and justification, and surely we as a community strive for future models that have both (thus understanding). However, our work clearly shows how any current approach to LLM training will actually fail because of exactly that. Since large-scale models have spread at an incredible rate not just through the AI/ML community but also to the industry and laymen, it is important to discuss safety critical settings, biases and presumptions. Our work intends on having a positive contribution to this by providing explanations and laying ground for a healthy discussion amongst peers of what these models are and are not capable of and how we ought to improve them.

**Funding.** The authors acknowledge the support of the German Science Foundation (DFG) project "Causality, Argumentation, and Machine Learning" (CAML2, KE 1686/3-2) of the SPP 1999 "Robust Argumentation Machines" (RATIO). This work was supported by the ICT-48 Network of AI Research Excellence Center "TAILOR" (EU Horizon 2020, GA No 952215), the Nexplore Collaboration Lab "AI in Construction" (AICO), the German Center for Artificial Intelligence (DFKI) and by the Federal Ministry of Education and Research (BMBF; project "PlexPlain", FKZ 01IS19081). It benefited from the Hessian research priority programme LOEWE within the project WhiteBox and the HMWK cluster project "The Third Wave of AI" (3AI).

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

## A  Appendix

This appendix provides optional, additional material the reader might find interesting. For reproduction of the the empirical part, our code is publicly available: https://github.com/MoritzWillig/causalParrots/

### A.1  Another Philosophical Argument in Favor of Meta SCM: "Self-referencing Systems"

As we have seen in our formalization, the Meta SCM idea seems to allow for variables that in some sense *talks about data generating processes themselves.* This is reminiscent of self-referencing systems that lie at the core of seminal arguments dating back to the origins of computer science. See for instance Turing's Halting problem proof (Turing et al., 1936) or Gödel's incompleteness proofs (Gödel, 1931). Essentially, we intend on asking a *philosophically* fundamental question that (as we have shown) implies other interesting questions of practical interest to the AI/ML community. Namely, to which extent does *understanding* causality differ from *knowing* causality? Such a question is certainly reminiscent of the Chinese Room Argument by Searle (2009). Therefore, if one could blur 'understanding' and 'knowing' causality, then this would imply that foundation models are causal because of the 'knowing' part but that effectively we could not tell the difference when only observing *output behavior.* Independent of the philosophical question–which by the way is beyond AI/ML systems an unresolved question also of human cognition–knowing to which extent we can rely on our foundation models to simply know the right causal answer for a causal query has important implications in AI/ML. The foundation model could be used (i) to head start learning with rough estimates, (ii) could serve as a recognition system for hidden variables that would require an increased computational complexity, and (iii) be used as interactive modules with human-in-the-loop.

## B  Technical Details and Metrics

The results in our paper were created on one NVIDIA A100-SXM4-80GB GPU with 80 GB of RAM and it takes 40 GPU minutes to query the OPT model. For the Luminous and GPT-3, we use the provided APIs, respectively.

### B.1  Sentence templates

Throughout our experiments we query the LLM about the presence or absence of a causal relationship between two variables within a data set. We represent our query in natural language form to the LLM using the following sentence templates, replacing $X$ and $Y$ with the respective variable names:

1. "Are $X$ and $Y$ causally related?"
2. "Is there a causal connection between $X$ and $Y$?"
3. "Is there a causality between $X$ and $Y$?"
4. "Does $X$ cause $Y$?"
5. "Does $X$ influence $Y$?"

It is worth noting that the sentence templates are *not equivalent*, some of them depict a difference in symmetry (e.g. (4) is clearly asymmetric), whereas others do not talk about causality (e.g. (5)). We pay careful attention to these distinctions while performing our empirical analysis.

## B.2 Definition of Sparsity

We define the Sparsity of a given graph as the number of present edges in relation to the maximum number of edges of the fully connected graph. For our evaluation we consider individual half edges, such that A → B and B → A are counted separately.

$$\text{Sparsity}(N, E_{\text{predicted}}) := 1 - \frac{\|E_{\text{predicted}}\|}{2 \cdot \binom{N}{2}}$$

where $N$ is the number of variables in the data set and $\|E_{\text{predicted}}\|$ is the number of actual edges in the graphs.

## B.3 Algorithm for ADS

We introduced a new metric for measuring the 'decisiveness' for an LLM when switching between asking asymmetric and symmetric queries. In other words, when we query a LLM with "Does $X$ cause $Y$?", then we are asking a question about X → Y and to our surprise we have observed that LLMs *always* decide more when asked in such way (i.e., we end up with more uni-directed edges). Thus, motivating the definition of such a new metric. The below algorithm presents the computation of Decisiveness:

---
**Algorithm 1** Decisiveness of a (causal) graph prediction

---
1: **procedure** DECISIVENESS(Predicted *Edges*)
2:     $sym \leftarrow 0$
3:     $asym \leftarrow 0$
4:     **for** *Edge* in predicted *Edges* **do**
5:         **if** *Edge* is directed **then**
6:             $asym \leftarrow asym + 1$
7:         **else if** *Edge* is symmetric **then**
8:             $sym \leftarrow sym + 1$
9:         **end if**
10:     **end for**
11:     $count \leftarrow asym + sym$
12:     **if** $count = 0$ **then**
13:         **return** $0.0$
14:     **else**
15:         **return** $asym/count$
16:     **end if**
17: **end procedure**

---

# C    Experimental Results

Additional material for all three main experimental sections from the main paper that could not be placed there due to space restrictions.

## C.1    How to Interpret LLM Output

Interestingly, both Luminous and OPT either fail to provide sensible answers or provide answers that are ambiguous, for instance, the LLM might *loop* indefinitely (repeating the first predicted sentence over and over again) or it might produce a "multiple-choice quiz" like output for which it will also choose an answer itself. GPT-3 was well behaved, providing sensible output throughout.

## C.2    Intuitive Physics

All the questions and answers for intuitive physics questions can be found at: <https://github.com/MoritzWillig/causalParrots/blob/master/media/intuitive_physics.pdf> In the following, we show two example interactions with the LLMs on the intuitive physics setting. First, we present the question which is input to the LLM and then we show for each of the three LLMs under consideration the respective answers/output. The markings 'continued' and 'repeating' mark that we have cut 'irrelevant' parts of the answer and the aforementioned 'looping' of the answer, respectively.

| | |
|---|---|
| "There is a tilted board above a bucket. Where does a ball end up if it is placed on the board?" | |
| GPT-4 | The ball will end up in the bucket. |
| GPT-3 | The ball will end up in the bucket. |
| Luminous | A: |
| | The ball will roll down the board. |
| | *[continued]* |
| OPT | A. It will fall into the bucket. |
| | B. It will fall into the bucket. |
| | *[continued]* |

| | |
|---|---|
| " ['A kilogram of metal is heavier than a kilogram of feathers'] is what most people say, but in reality" | |
| GPT-4 | a kilogram of metal weighs the same as a kilogram of feathers. The weight is the same because a kilogram is a kilogram regardless of the material being weighed. The difference lies in the volume or space that each material occupies. Metal is *[end of output]* |
| GPT-3 | they weigh the same. |
| Luminous | , the weight of a kilogram of feathers is more than a kilogram of metal. |
| | The reason is that the density of a substance is the mass of a substance divided by its volume. |
| | *[continued]* |
| OPT | , a kilogram of feathers is heavier than a kilogram of metal. |
| | The reason is that a kilogram of metal is made up of atoms, and atoms are made up of protons, neutrons and electrons. |
| | *[repeating]* |

### C.3 Propositional Logic

All the questions and answers for the basic propositional logic (causal chains) questions can be found at: https://github.com/MoritzWillig/causalParrots/blob/master/media/causal_chains.pdf It is the equivalent setup to the previous subsection just as the analogue for the propositional logic.

| "If A causes B and B causes C. Does A cause C?" | |
|---|---|
| GPT-4 | Yes, in a sense, A does cause C indirectly through B. This is known as a causal chain. However, the relationship may not be as strong or direct as the relationship between A and B or B and C. Other factors could also influence *[end of output]* |
| GPT-3 | It is possible that A causes C, but it is also possible that A and C are unrelated. |
| Luminous | A: |
| | The answer is no. |
| | *[continued]* |
| OPT | If A causes B and B causes C, then A causes C. |
| | If A causes B and B causes C, then A does not cause C. |
| | *[continued]* |

| "If A causes B and B causes C. Does A cause B?" | |
|---|---|
| GPT-4 | Yes, as per the initial statement, A causes B. |
| GPT-3 | There is not enough information to answer this question. |
| Luminous | A: |
| | The answer is no. |
| | *[continued]* |
| OPT | If A causes B and B causes C, then A causes C. |
| | If A causes B and B does not cause C, then A does not cause C. |
| | *[repeating]* |

### C.4 Changing the Variable Names for Prediction

The following is an additional experiment for the graph prediction setting from the main paper where we are given (partially) the ground truth causal graph. We changed the variable names to synonyms (or at least arguably similar concepts) and observed how *variable renaming might induce unsmooth change*. In this setting, we consider the single graph of the Health data set, which involves the variables 'age', 'nutrition', 'health', and 'mobility' and alternate the variable names (see following figure). We either choose words recognized as *synonyms* of the original variable wording or words that might appear in a similar context but have an identifiable difference to the original word. What you see in the figure is three rows (one for each of the LLMs) of graph prediction differences (difference to the original prediction denoted as the change for any single edge prediction) based on our standard setting of querying with different causal question templates, only on the Health data set, for the different variable naming alternatives. A single key observation was made. Luminous reacted with increased sparsity in graph prediction when changing the variable 'mobility' to mean 'fitness'. On the other hand, GPT-3 conversely reacted with decreased sparsity in graph prediction when changing the variable 'age' to 'aging.' Arguably, the former change is more drastic than the second since fitness as a concept might refer to a superset that includes mobility but also other things like conditioning etc., whereas aging solely refers to the process of increasing the age. The pattern seems overall arbitrary, but we believe the observation that 'similar' words might cause drastic change is noteworthy.

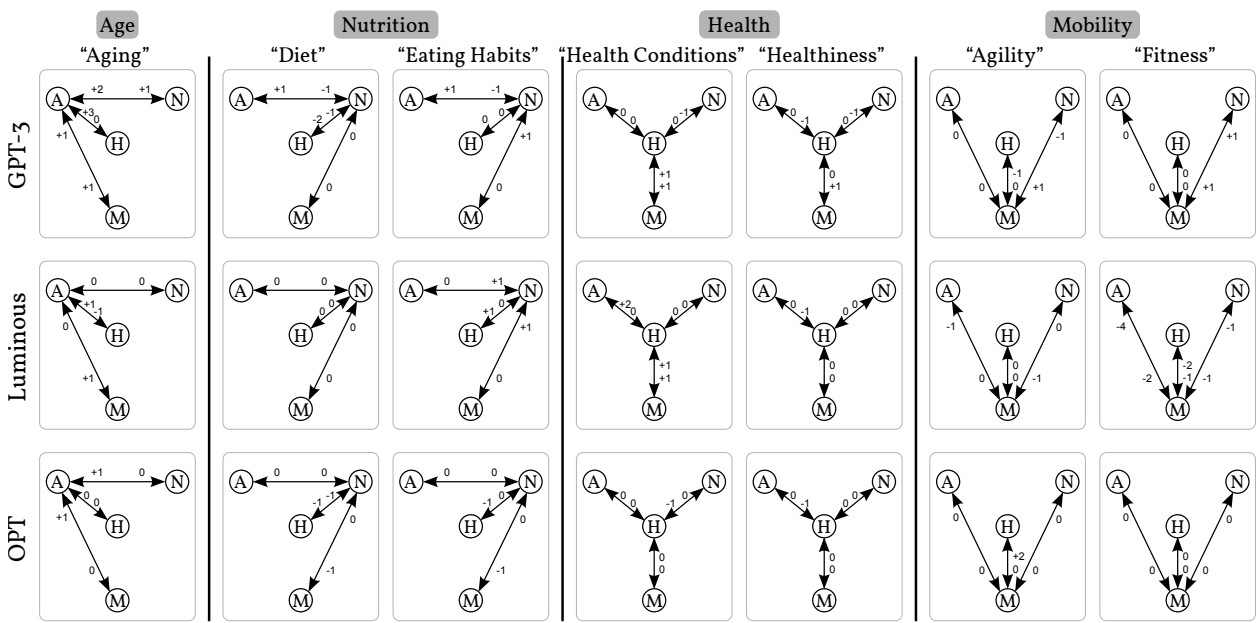

Figure 6: **Sensitivity to the Naming of the Variable Concepts.**

## C.5 Chain of Thoughts Prompting

Within several experiments of this paper Chain of Thoughts (CoT) prompting is deployed. In consequence a number of exemplary question answer pairs are placed before the actual input. These QA pairs match the style of the task of interest, but use an exclusively distinct set of variables. Upon evaluation the first $N$ QA pairs from the following lists where presented to the model. The following QA pairs where used for the Causal Chain experiments:

1. Q: If X causes Y and Y causes Z. Does X cause Z?
   A: Because X causes Y and Y causes Z, X causes Z. The answer is yes.
2. Q: If Z causes Y and Y causes X. Does Z cause X?
   A: Because Z causes Y and Y causes X, Z causes X. The answer is yes.
3. Q: If X causes Y and Y causes Z. Does Y cause X?
   A: Because Y does not cause X directly and Y causes Z which does not cause X, Y does not cause X. The answer is no.
4. Q: If Y causes Z and X causes Y. Does X cause Z?
   A: Because X causes Y and Y causes Z, X causes Z. The answer is yes.
5. Q: If Y causes Z, W causes X, X causes Y and V causes W. Does M cause Z?
   A: Because M does not appear in any of the clauses, the answer is no.
6. Q: If Y causes Z, W causes X, X causes Y and V causes W. Does V cause Z?
   A: Because V causes W, W causes X, X causes Y and Y causes Z, the answer is yes.
7. Q: If V causes W, W causes X, X causes Y and Y causes Z. Does X cause W?
   A: Because X only causes Y and Y only causes Z, there is no directed path from X to W. The answer is no.
8. Q: If Y causes Z, W causes X and V causes W. Does V cause Z?
   A: V causes W and W causes X, but neither V, W nor X cause Z. The answer is no.

The following QA pairs where used for the Natural Word Chain experiments.

1. Q: If flipping switches causes light bulbs to shine and shining light bulbs cause moths to appear. Does flipping switches causes moths to appear?

| Natural Word Chain (15 prompts) | | | | | | |
|---|---|---|---|---|---|
| CoT Length | plain prompting | 1 | 2 | 3 | 4 |
| GPT-4 | 13 | 14 | 13 | **15** | **15** |
| GPT-3 | 5 | 10 | 11 | 10 | 9 |
| Luminous | 5 | 10 | 11 | 10 | 9 |
| OPT | 4 | 10 | 8 | 10 | 5 |

| Causal Chains (20 prompts) | | | | | | | | | |
|---|---|---|---|---|---|---|---|---|
| CoT Length | plain prompting | 1 | 2 | 3 | 4 | 5 | 6 | 7 | 8 |
| GPT-4 | **20** | - | - | - | - | - | - | - | - |
| GPT-3 | 9 | 17 | 17 | 19 | **20** | 19 | **20** | 19 | 19 |
| Luminous | 10 | 15 | 14 | 11 | 12 | 6 | 7 | 7 | 6 |
| OPT | 4 | 6 | 15 | 14 | 16 | 6 | 8 | 7 | 8 |

Figure 7: Results for Chain of Thoughts prompting experiments. Numbers indicate the total number of correctly answered questions. Luminous and OPT seem to benefit most from lower numbers of provided examples compared to GPT models.

    A: Because flipping switches causes light bulbs to shine and shining light bulbs causes moths to appear, flipping switches causes moths to appear. The answer is yes.

2. Q: If heavy rain causes the streets to flood and shining light bulbs cause moths to appear. Does heavy rain cause moths to appear?
   A: Because heavy rain causes the streets to flood, but is not connected to shining light bulbs which allow for moths to appear. The answer is no.

3. Q: If Sibfan causes flooded streets and heavy rain causes Sibfan. Does heavy rain cause flooded streets?
   A: Because heavy rain causes Sibfan and Sibfan causes flooded streets. The answer is yes.

4. Q: If Sibfan causes heavy rain and Sibfan causes light bulbs to shine. Does Sibfan cause heavy rain?
   A: Because Sibfan directly causes heavy rain, the answer is yes.

Fig. 7 lists the absolute number of correctly answered queries for all CoT experiments conducted in this paper. Individual answers to all queries can be found under `https://github.com/MoritzWillig/causalParrots/tree/master/media`.

### C.6 Embedding predictions

Graph predictions of GPT-3 embeddings of the ConceptNet knowledge graph facts. We embed causal and anti-causal facts of the ConceptNet data set to gain a set of 'labeled' causal embeddings. To predict an edge of a data set, we instantiate the query text using our five sentence templates. We embed the queries and perform a k-NN search, comparing the query embedding to all ConceptNet facts. The presence of an edge is decided based on the label with the most similar embedding (measured in terms of correlation which in this case amounts to cosine similarity). That is, we decide for the edge to be present if the most similar embedding stems from a causal fact of the ConceptNet-based data set. If the nearest neighbour stems from an anti-causal fact, we predict the edge to be absent. In the following you see for each of the six data sets and each of the five query alternatives the respective graph prediction.

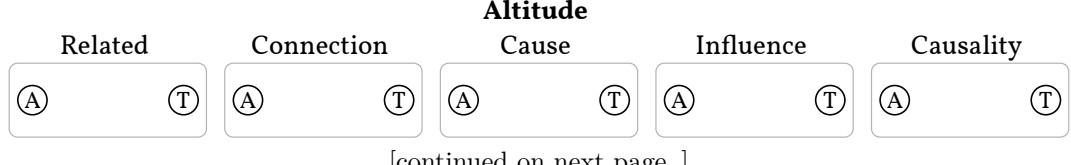

[continued on next page..]

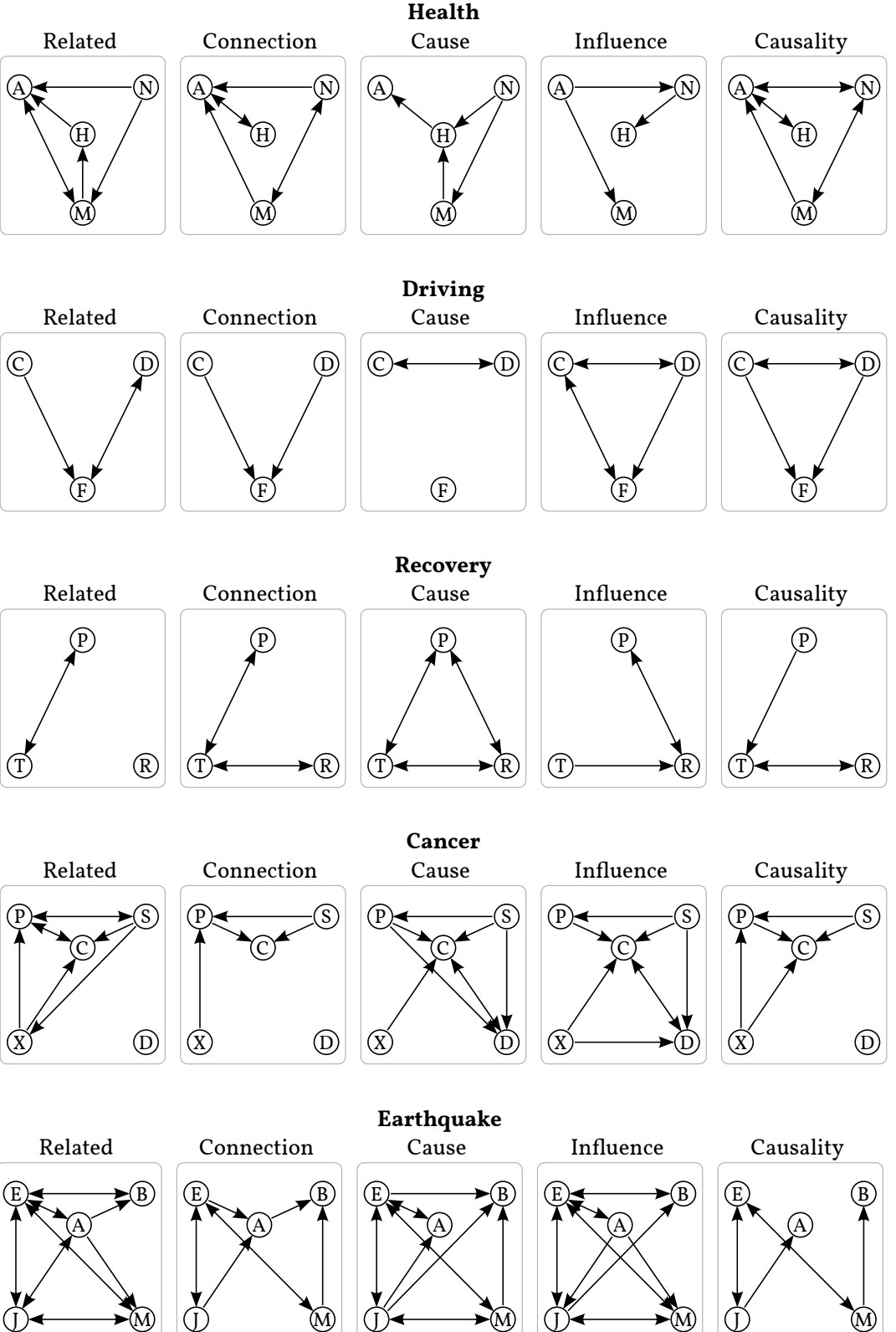

Figure 8: **Graph Predictions Based on the Knowledge Base Fact Embeddings (k-NN).** For each of the data sets and each of the query sentence templates.

