# OpenReview forum: "Causal Parrots: Large Language Models May Talk Causality But Are Not Causal"
_TMLR — Accepted by TMLR_

### Review · Reviewer_WdpA · 2023-05-26

**Summary Of Contributions:**

This paper focuses on the public discussion around the belief that scaling is all that's needed to achieve artificial general intelligence, especially with respect to causal modeling capabilities. The authors strongly posit that large language models (LLMs) are incapable of genuinely understanding causal relationships, despite appearances suggesting otherwise.

The key contribution of this paper is the introduction of a novel subcategory of Structural Causal Models (SCMs) known as 'meta SCMs', which have the ability to encode causal information about other SCMs within their own variables. The paper presents a hypothesis that in cases where LLMs appear to be successful in performing causal inference, this success is likely because of a meta SCM lying beneath the surface. This underlying model reveals correlations between causal facts in natural language, and this data is ultimately what the LLMs are trained on.

An analogy used in the paper to describe this phenomenon likens LLMs to parrots, simply repeating the causal knowledge already present in the data they were trained on, without truly understanding it. Through empirical analysis, the researchers try to provide evidence supporting their theory, suggesting that LLMs might not even be particularly efficient 'causal parrots'.


**Audience:**

Yes

**Broader Impact Concerns:**

I have no concerns about the ethical implications of this work that would require adding to a Broader Impact Statement.



**Claims And Evidence:**

No

**Requested Changes:**

As mentioned in the weaknesses section, my main concerns are the lack of empirical support, the missing literature and the unsubstantiated logical leaps.

Therefore, I’m requesting two main changes: (**a**) that the authors provide additional experiments that actually support their claims, where LLMs are clearly shown to fail because of their memorization of causal facts, while humans succeed. (**b**) that the authors rewrite the paper in a manner that supports a scientific discussion of the claims, removing all unsubstantiated claims and cleaning up the phrasing in a way that strong arguments are supported by strong evidence.

Apart from that, to address the missing literature, I’m adding here a list of some potentially relevant papers that I recommend the authors address (this is a very small but representative sample):

1. Causal Inference in Natural Language Processing: Estimation, Prediction, Interpretation and Beyond

2. Invariant risk minimization

3. Invariant causal prediction for nonlinear models

4. Causal abstractions of neural networks

5. Causalm: Causal model explanation through counterfactual language models


In addressing all of this body of work, I believe that the analysis and evaluation would be much more useful to the TMLR audience. If the authors seriously engage with this literature it will strengthen this work substantially.


**Strengths And Weaknesses:**

### Strengths:

The paper discusses an important and timely problem - probing the causal understanding of large language models. It is one of the most intriguing scientific debates in recent history, and the authors provide a theoretical argument for what LLMs are capable of in terms of understanding the natural world. Concretely, their causal parrots arguments, which demonstrated how models can mimic causal knowledge by memorizing causal facts, is both interesting and important for the scientific discussion around LLMs. The meta-SCM is a novel idea that can be useful for this debate.

### Weaknesses:

While I whole-heartedly share with the authors the belief that understanding and defining the causal capabilities of LLMs is fascinating, I find the paper in its current form limited. To be concrete, here are the main weaknesses of the current version of the paper in my opinion:

1. Problem motivation and set up - I find the description of the motivation for the paper, especially in the introduction, unconvincing. While I do agree that it is interesting and worthwhile to understand whether LLMs “talk” causality, I don’t agree with the logical steps in the introduction, setting this up as the crucial point of disagreement in the AGI debate.

2. Writing//language -the topic being discussed in the paper is incredibly popular, and many people have expressed their opinion publicly about what LLMs are capable of doing. However, I think that to be accepted to TMLR, the language should be substantially more scientific, and adhere to more formal statements. This could be improved easily in the next iteration of the paper, if authors edit their text. As it currently stands, the paper reads like a blog post, making many unfounded claims about causality, NLP and ML.

3. Gap between strength of claims and weakness of empirical results - results in this paper and in Kiciman et al 2023 demonstrate the strength of LLMs, not their weakness (and I imagine that with GPT4 the results would be even more positive for LLMs). I don’t see how this does not affect the very strong claims made in the paper suggesting that models are “causal parrots”. I’d expect for the type of claims made in the paper to see empirical results that compare LLMs to humans, or other challenging comparisons that prove the claims. What I’m seeing in the paper are strong theoretical claims suggesting that models can’t understand higher rungs of causality, but no empirical results demonstrating clear failure modes.

4. Complete disregard of the literature on Causal Inference+NLP.
While the authors dive into the broader AGI and causality discussion, they completely ignore the literature that studies how to use tools from causal inference to improve NLP models, including causal explanations, invariant learning and causal abstractions of LMs.


### Minor Issues:

There are many writing errors that need to be addressed. I highly recommend this paper goes through a round of editing to change both grammar and writing flow. I personally dislike the style of writing this paper as an informal online debate, but I also respect the authors choice, and as long as claims are substantiated it’s their choice. Here are a few grammar/language error to provide some reference:

**Abstract**: “the cases *were LLM” -  *where

**Introduction**: The writing in the intro is a bit odd. The authors include an entire block of text without a proper citation, all at the expense of actually introducing what they are going to show in the paper. I don’t know whether the authors wrote the block or if it’s taken out of some public debate, but in any case I didn’t find it useful for setting up the paper.

Also, the second to last paragraph includes a logical leap, suggesting that understanding whether current LLMs “talk” causality is the key to settling this debate. I didn’t follow the logical steps getting the authors to this conclusion.

**Examples 1/2** - Consider not using the definition environment, there’s a lot of text and it’s harder to read it with italics.

**Section 6** - “As we started exploring in Sec.3, nature ultimately any sort of causal assumptions that we can talk about.” - this sentence is unclear.

---

> ### Author Response · Authors · 2023-07-07
> **Answer to Reviewer WdpA**
>
> We thank reviewer WdpA for the detailed comments/questions/observations in "requested changes"/weaknesses but we also wish to constructively/kindly push back on some of these comments in the following argumentation, in hope of converging to a better paper together.
>
> Regarding the motivational aspect and the "crucial point of AGI debate" aspect: we'd love to hear your precise opinion on why you don't agree with our motivation, as of your current review we can only perform guesses since you have not listed your reasons. Even if you did not have any particular reasons (in the sense that it is simply a hunch that the LLM discussion is not a crucial part), it is important to not forget that we claim that LLMs are the "central question" to AGI. The only thing we conclude is that it has been the catalyst for the heated discussions between researchers. Putting conflicts of interests (like for companies that profit from the success of LLMs e.g. Microsoft to mention but an example) aside, the success stories written by LLM applications are undeniable, however, their implications whether it is "true sparks" of AGI are clearly debatable. In essence, this is the motivation for our work because we intend on scientifically investigating what is true in said debate/discussions/speculations.
>
> Regarding the second point, which is really just part of the first bullet, we could of course have a more general debate about what should be considered "scientific" writing at this point. We are well aware that this initial section reads very different from any typical paper, however, we've made this decision on purpose (for all the reasons mentioned above as well). Furthermore, it is not like we leave the reader unprotected from this fact, quite to the contrary we make use of special formatting and make sure to reference accordingly within our summary of events/opinions that happen outside conventional academic literature. These aspects are furthermore something we've already added to the paper as a careful consideration and followup to the prior work this present works builds upon. Again, this is really our only "push back" point, and we'd love to hear your opinion of how our two differing opinions on this could converge to something that you would find beneficial/acceptable (since arguably a lot of other researchers might share the same view as you). Maybe by having an even more explicit disclaimer? Or do you believe in no way this is acceptable and we should rewrite the Introduction and push the current discussion to the Appendix?
>
> Regarding the strength of the empirical results: following suggestions by your fellow reviewers, we've added new experiments for GPT-4 but also new experiment setups such as an improved propositional setup and more recent approaches from NLP literature such as Chain-of-Thought prompting (which is what you touch upon in your last bullet). Please consider these results as they give a more thorough view. Nonetheless, the CCF conjecture still (to our unfortune) remains a conjecture. While especially the GPT-4 result seems to maybe provide another hint towards the conjecture being true considering the "meta SCM and fine-tuning" discussion that we've added to the paper, it is still no definite claim. Regarding the paper by Kiciman et al 2023 that you specifically mention, we have two comments on this. (1) the paper actually references a prior version of the present work, which acknowledges our results (which have been strengthened a lot more with the followup work and even during this rebuttal) and places it into the context of their own work. In fact, the last boldface paragraph of their paper is dedicated to a discussion with our paper. (2) one big discussion in their paper are the positive results they observe for the Tübingen bivariate causal data set from Mooij et al. 2016. While impressive at first glance maybe, looking at the actual data set renders the result really lacking. The data set can be found here: https://webdav.tuebingen.mpg.de/cause-effect/. The authors actually state themselves "However, we do not guarantee that all provided ground truths are correct." Therefore, we have 108 (X,Y) pairs for which we don't necessarily have ground truth. Like in our setup, the actual data measurements are not used, but only the concepts like for pair 0001 which is X: altitude and Y: temperature. This very same example we employ as well and is arguably reasonable since these are "common sense"/every-day concepts. But what about example pair 0085 for instance where X: time to take weekly measurements (from 1 to 14) and Y: protein content of the milk produced by each cow at time X? This example is not cherry-picked and yet it becomes very apparent that the then "good" LLM prediction results (by the way measured simply in terms of hit-and-miss accuracy) are meaningless. In conclusion, both (1) and (2) highlight that the overall discussion is nuanced and that Kiciman et al 2023 are in favor of our paper's results

---

> > ### Author Response · Authors · 2023-07-07
> >
> > Regarding the human argument  we carried Chain of Thoughts prompting experiments as requested by the other authors. In the prompts we presented up to eight exemplary question answer pairs before appending the actual question of interest. While this style of prompting generally helped models to greatly improve their performance, we also wanted to note that CoT prompting intrinsically makes it difficult to investigate the CCF hypothesis. More specifically, the models are provided with an exemplary ‘thought process’ to solve a given task. By providing these 'chains of thoughts' we 'materialise' the otherwise unobserved thought process and provide it via the means of input data. As LLMs are considered few-shot learners, we are no longer able to distinguish between the models’ understanding of the task and the model simply mirroring the provided thought process. Real understanding of the problem, however, can only be tested by the models solving a problem via induction. This would require the model to take the description of a task and utilize the given information towards solving it. As no examples are allowed to be provided, this rules out CoT style querying and in turn decreases performance. LLMs might be trained or fine-tuned to inductively reason about arbitrary problems. While this provides an interesting research direction, we feel that this sort of training is out of scope for this paper. While such behaviour can be interpreted as a flaw LLMs, we also want to point out that humans sometimes adopt similar behaviour by carrying out a certain sequence of actions to solve a task. For example, children may solve certain maths problems by applying steps of an algorithm rather `mechanically' than understanding the actual implications of these steps towards producing the right answer.
> >
> > Regarding the discussion of causality+NLP literature. We've added more discussion and references now to the paper. However, the 3 papers that you explicitly mention as "representative" being IRM, ICP and Causal Abstractions of NNs are almost completely unrelated to our work (they are not entirely unrelated since they consider causality), also they most certainly are not works in the causality+NLP realm.
> >
> > Thanks for highlighting in detail some of our writing mistakes. We've corrected them and also done a third-party proof reading pass.
> >
> > Thank you again, we hope to have convinced you that with the new changes our paper is ready for publication with TMLR and we look forward to further discussions.
> >
> > Your authors
> >
> > p.s. we've incorporated all the requested changes from your end, **your color code is blue**.

---

### Review · Reviewer_wg1g · 2023-06-15

**Summary Of Contributions:**

This paper presents a comprehensive empirical study that explores the extent to which large language models (LLMs) can effectively answer causal questions. The study investigates the causal capabilities of LLMs across three distinct causal settings: 1) commonsense reasoning, 2) causal discovery, and 3) augmenting with additional knowledge. To conduct the experiments, the researchers selected three recent LLMs, namely GPT-3, Luminous, and OPT. The findings of the study indicate that LLMs do not possess inherent causal modeling abilities and are limited in their capacity to provide accurate answers to causal questions.

**Audience:**

Yes

**Broader Impact Concerns:**

To the best of my knowledge, I do not see any ethical concerns.

**Claims And Evidence:**

Yes

**Requested Changes:**

1. Could you conduct another experiment for the "propositional chain", in which using concrete variables instead of symbolic letters. In this way, it can be guaranteed that we are focusing on the reasoning of the "propositional chain".

2. Could you design stronger prompts using the "Chain of Thoughts (CoT)" strategy, and evaluate the LLM's causal discovery capability? I believe this will make the conclusion more convincing. Although I mentioned "Tree of Thoughts (ToT)" in the weakness section, it is not necessary to try ToT since it is quite new and needs further understanding.

**Strengths And Weaknesses:**

**Strengths:**

- The paper addresses a timely and important topic by evaluating the causal reasoning ability of Large Language Models (LLMs), which have shown promising performance on various tasks and are of interest in the context of Artificial General Intelligence (AGI).

- The three settings explored in the study (commonsense reasoning, causal graph discovery, and including additional knowledge) provide a comprehensive evaluation of the LLMs' causal reasoning abilities, covering different aspects of causal reasoning.

- The experiments conducted in the study are detailed and provide valuable insights into the LLMs' performance on causal reasoning tasks. Factors such as sensitivity to variable concept names and the length of causal chains are taken into account, contributing to a better understanding of how LLMs operate in the context of causal reasoning.

Overall, these strengths highlight the significance of the paper's contribution and the thoroughness of the evaluation conducted, making it a valuable addition to the existing literature on LLMs and their causal reasoning abilities.


**Weaknesses:**

The major limitation of this study is the inappropriate prompts used as input for LLMs.

In the commonsense inference task, the authors use a "propositional chain" prompt format, namely "X_1 causes X_2 ... and X_{n−1} causes X_n". However, LLMs are known to struggle with symbolic reasoning tasks when provided with simple prompts alone [1]. Therefore, this experiment alone may not be sufficient to assess the capability of LLMs in "propositional chain" reasoning.

Although the paper includes various templates of prompts (queries) such as those in Section B.1, these prompts are generally simple and plain. Recent studies indicate that such basic prompts may not fully unlock the strong reasoning abilities of LLMs [1]. These studies propose stronger prompt strategies, such as "Chain of Thoughts (CoT)" and the more recent "Tree of Thoughts" [2], which notably enhance the reasoning capabilities of current LLMs. Therefore, relying solely on simple prompts may not be convincing enough to draw definitive conclusions.

One advantage of LLMs is their ability to serve as base models and answer questions from diverse domains. However, it should be acknowledged that SCM (Structural Causal Models) is highly specialized for a specific group of variables. If we were to perform instruction tuning or fine-tuning using specialized data for SCM, could LLMs effectively answer causal questions in that particular domain?



[1]  Wei, Jason, et al. "Chain of thought prompting elicits reasoning in large language models." arXiv preprint arXiv:2201.11903 (2022).
[2] Yao, Shunyu, et al. "Tree of thoughts: Deliberate problem solving with large language models." arXiv preprint arXiv:2305.10601 (2023).

---

> ### Author Response · Authors · 2023-07-07
> **Answer to Reviewer wg1g**
>
> We thank the reviewer wg1g for the detailed comments/questions/observations in the "requested changes"/weaknesses section which helps us improving our paper.
>
> Before discussing the changes, we just wanted to point out (just like your fellow reviewer WdpA pointed out in their review) that a key contribution of ours (possibly even the main one!) lies in the formalism of meta SCM and the accompanying CCF conjecture, which was left out in your review. Given that you've found appeal in our empirical study, we do think that you would also prefer our theoretical idea that justifies and kickstarts our empirical investigation (in hope of revealing parts of the conjecture).
>
> Regarding the prompting techniques employed in the commonsense and causal discovery categories of the evaluation, we've now followed your advice and performed evaluations with an improved propositional reasoning approach and also Chain-of-Thought prompting. Furthermore, following the suggestions by fellow reviewer QJjp, we've added experiments with GPT-4. You can find the new results in the paper. In summary,
>
> Symbolic variable names: We are sorry about the unclear formulation in our text and have added a clarifying statement. Upon instantiation of the queries, placeholder variables $X_i$ are replaced with corresponding letters: X_1 -> 'A', X_2 -> 'B',... Nonetheless, we wanted to investigate this issue further. As such, we performed a small evaluation regarding symbolic and non-natural variable names. Results are added to Table 1 and are discussed in Sec. 4.1. We found no significant difference in causal inference capabilities of the models when switching between real world, imaginary or symbolic variable names.
>
> As requested, we carried Chain of Thoughts prompting experiments. In the prompts we presented up to eight exemplary question answer pairs before appending the actual question of interest. While this style of prompting generally helped models to greatly improve their performance, we also wanted to note that CoT prompting intrinsically makes it harder to investigate the CCF hypothesis. More specifically, the models are provided with an exemplary ‘thought process’ to solve a given task. By providing these 'chains of thoughts' we 'materialise' the otherwise unobserved thought process and provide it via the means of input data. As LLMs are considered few-shot learners, we are no longer able to distinguish between the models’ understanding of the task and the model simply mirroring the provided thought process. Real understanding of the problem, however, can only be tested by the models solving a problem via induction. This would require the model to take the description of a task and utilize the given information towards solving it. As no examples are allowed to be provided, this rules out CoT style querying and in turn decreases performance. LLMs might be trained or fine-tuned to reason inductively about arbitrary problems. While this provides an interesting research direction, we feel that this sort of analysis is out of scope for this paper.
>
> We look forward to discussing with you, thanks.
>
> Your authors
>
> p.s. we've incorporated all the requested changes from your end, **your color code is purple**.

---

> > ### Comment · Reviewer_wg1g · 2023-07-27
> >
> > Thanks for the authors' response. My concerns are well addressed.

---

### Review · Reviewer_QJjp · 2023-06-26

**Summary Of Contributions:**

This work seeks to answer a fundamental question about large language models, which is whether they are able to serve as causal reasoning agents. The authors put forward an argument by defining a "meta" causal model which can answer interventional questions about another, distinct, structural causal model. The authors then point out that while these meta models inherit causal semantics, they are not necessarily reliable or reasonable causal models for the physical world. This is an important argument, since it mirrors the setting of LLMs which learn over training data that contains textual description of learnt causal knowledge rather than measurements. The authors the put forth a conjecture that correlating these causal facts will minimize training error, implying that they are able to produce causal inferences as a consequence of those facts being present within the training corpus. The authors then test this conjecture empirically on three prominent LLMs. The empirical evidence largely mirrors the presented conjectures, however the authors also note occasions where the produced answers do appear to mimic human reasoning.

**Audience:**

Yes

**Broader Impact Concerns:**

This area is an incredibly sensitive one and I appreciate the authors' broader impact statement.

**Claims And Evidence:**

Yes

**Requested Changes:**

* As noted above, it would be nice to have a slightly more nuanced discussion of LLMs in terms of fine tuning for specific domains.
* I think it is also important to acknowledge that some of these findings may be impacted by the release of more powerful models such as GPT-4 (though I believe the results and overarching framework are still very much relevant and important to the community)
* There are a number of typos and grammatical errors (e.g., "countries's") that should be addressed with a careful editing pass.

**Strengths And Weaknesses:**

This work takes a principled, sober, and reasonably thorough approach to evaluating the effectiveness of LLMs as agents of causal reasoning. I think the formalism and accompanying conjecture put forth by the authors is a reasonable one and the accompanying experiments are a good set of first steps toward understanding the behavior of LLMs in the context of causal reasoning. With that being said, the paper is limited in scope in the sense that it is attempting to reason over fully general LLMs like GPT. While understandably out of scope for this work, it is an important question to ask when the meta-SCM provides a reliable proxy for causal knowledge, and whether careful fine tuning targeted for domains of interest can serve as such proxies. It would be useful to acknowledge this within the larger discussion of the work. It also is a little unfortunate that the LLMs considered don't include more recent models such as GPT-4, but this in my view does not warrant rejection.

---

> ### Author Response · Authors · 2023-07-07
> **Answer to Reviewer QJjp**
>
> We thank the reviewer QJjp for the detailed comments/questions/observations in the "requested changes" section which helps us improving our paper.
>
> Regarding the question on when meta SCM are "reliable." Indeed, as you pointed out yourself, the question might be out of scope for this particular work, still, it is clearly related and can be considered as a concrete followup direction to the present work. Fine-tuning is indeed the way to go for many applications, as discussed in the "foundations model" paper from Stanford, and equating this to meta SCM is surely intriguing! As of now, we only have some guesses. We've added a new extended discussion to the paper.
>
> Regarding GPT-4, we've re-run our emprical evaluation with this newer version of GPT under careful consideration of the setup to ensure a fair evaluation. In short: GPT-4 performs better in the "common sense" category, whereas performing just "as bad" in the "specific ground truth" category of the experiments. This is in the realm of our prior expectations, however, it is worthwhile to note that in the former category GPT-4 performing well might in fact be due to our data being part of GPT-4's training data, since a prior workshop paper version of this work was already made publically available on arXiv and other conference proceedings, thus being available at the time of GPT-4 training (unlike GPT-3 and all other models like OPT and Luminous previously evaluated). Unfortunately, we do not have a way of proving this conjecture.
>
> Thanks for pointing our grammar mistakes and such, we've done proof-reading now also with a third-party to avoid mistakes that the authors are simply unable to spot after staring on the same document for so long.
>
> We look forward to discussing with you, thanks.
>
> Your authors
>
> p.s. we've incorporated all the requested changes from your end, **your color code is orange**.

---

> > ### Comment · Reviewer_QJjp · 2023-07-14
> >
> > Thank you for your response, and your updates to the text. After reading, I think you've done a nice job of addressing my outstanding concerns.

---

### Author Response · Authors · 2023-07-08
**Revision with Requested Changes Complete**

Dear reviewers,

thank you again for the valuable feedback!

The revised paper has been uploaded (with color and also reviewer-specific color codes).

We look forward to further talking to you.

Best regards,
Your authors

---

### Decision · Action_Editors · 2023-08-17

**Recommendation:** Accept as is

**Comment:**


This paper tackles a great problem---how "aware" are large language models of causality? The authors set up a simple hypothesis: to the degree that LLMs are capable of producing causal outputs, it is the result of seeing such (or similar) causal relationships in their training data, rather than having learned underlying causal reasoning. There are two substantial contributions here: first, an overall framework to formalize the hypothesis, and second, a set of experiments testing the causal capabilities of current LLMs.

Generally, all reviewers agree that (1) the problem is timely and important, (2) that the authors provide thought-provoking ideas and content. All of the reviewers offered constructive feedback, and the authors have substantially strengthened the paper, including broader experiments (e.g., dealing with the typically brittle nature of prompts) and far more background.

The only downside of the paper is that it sets up a very challenging problem and can only make a set of empirical arguments in one direction. However, I do not believe this is a major limitation. My opinion is that the authors set up a nice initial work on an area that is bound to be vast (large pretrained models and their causal capabilities). The work is quite far from a last word on the subject, but serves to open up one part of the area and has several interesting insights.

For this reason I recommend acceptance.

**Audience:**

Yes, it is a very timely paper.

**Claims And Evidence:**

Yes, the level of evidence is sufficient.

---

> ### Author Response · Authors · 2023-08-22
> **Camera-Ready + Video + Code Online**
>
> Dear AE, Dear Reviewers,
>
> thank you once more for the great discussions that improved our paper to its now final version.
>
> The camera-ready version of the paper alongside a video / talk recording and the code for reproducing the empirical part are now all online.
>
> The links can be found in the listing above.
>
> Kind regards,
> your authors